# Adaptive Interest for Emphatic Reinforcement Learning

**Martin Klissarov***
Mila, McGill University

**Rasool Fakoor**
Amazon Web Services

**Jonas Mueller**
Cleanlab

**Kavosh Asadi**
Amazon Web Services

**Taesup Kim**
Seoul National University

**Alexander J. Smola**
Amazon Web Services

## Abstract

Emphatic algorithms have shown great promise in stabilizing and improving reinforcement learning by selectively emphasizing the update rule. Although the emphasis fundamentally depends on an interest function which defines the intrinsic importance of each state, most approaches simply adopt a uniform interest over all states (except where a hand-designed interest is possible based on domain knowledge). In this paper, we investigate adaptive methods that allow the interest function to dynamically vary over states and iterations. In particular, we leverage meta-gradients to automatically discover online an interest function that would accelerate the agent's learning process. Empirical evaluations on a wide range of environments show that adapting the interest is key to provide significant gains. Qualitative analysis indicates that the learned interest function emphasizes states of particular importance, such as bottlenecks, which can be especially useful in a transfer learning setting.

## 1 Introduction

A fundamental challenge in reinforcement learning (RL) is to approximate key quantities such as value functions and optimal policies. Under the assumption that the world in which an RL agent interacts is large and the computational capacity is limited, a natural trade-off emerges in which certain quantities are more accurately predicted than others over the course of learning. Standard RL algorithms, such as temporal differences (TD) [39], perform updates at every state, thereby spending more resources on such frequent states at the expense of other potentially more useful ones. A possible solution could be to selectively emphasize certain updates, for example through a state-dependent *interest function*. However, when combined with standard bootstrapping as in TD($\lambda$), such update rules are known to be unstable [25].

*Emphatic* algorithms propose a solution in which state-dependent selective updating can be applied while maintaining stability under linear function approximation [44]. At their core, emphatic algorithms determine the emphasis to be applied at each update by accounting for how much the current state is being bootstrapped from as well as an intrinsic measure of its importance relative to other states. This intrinsic measure is encoded through an arbitrary state-dependent *interest* function which the practitioner can set to any desired positive value. However, with this added flexibility comes a problematic question: how should one select the interest function?

Previous works propose hand-crafted solutions for the interest function that showcase the usefulness of selective updating [27, 3, 28]. However, when applying emphatic algorithms to complex environments

---

*Correspondence to: Martin [martin.klissarov@mail.mcgill.ca] and Rasool [fakoor@amazon.com].

36th Conference on Neural Information Processing Systems (NeurIPS 2022).

where external domain knowledge may be too hard to encode, other than rare and specific exceptions [59], most practitioners use a uniform interest over states [44, 53, 20, 60, 21, 17]. That is, they simply set the interest to 1 for all states. Building on the intuition that it can be beneficial to learn more from certain states than others, we argue that *different* emphases may be useful at various stages of the learning process. Indeed, as the RL learning process is inherently non-stationary, the relative importance of a particular state in the agent's updates should likely vary over training iterates as well.

In this work, we study how to adaptively learn the interest function in complex environments where hand-crafting an effective interest function is impractical. A good approach should allow for fast and flexible adaptation based on the agent's interactions with its environment. Considering the previous success of meta-gradient framework in discovering hyperparameters [55, 58], objective functions [54], intrinsic rewards [62], and temporal abstractions [51], we here propose to learn and adapt the interest function based on meta-gradients in an online fashion. The interest function in our method is parameterized by *meta-parameters*, which are updated by gradient descent along with the parameters of the policy and value function.

We empirically investigate the merits of adapting the interest function on a wide variety of environments and settings, ranging from prediction with linear function approximation to control on vision-based tasks. Our contributions are the following. (1) In the off-policy setting, we see substantial gains in performance and sample efficiency when adapting the interest function. (2) We extend the traditional application of emphatic algorithms from the off-policy setting to on-policy control, where we find it is crucial to adapt the interest function in order to observe consistent gains. (3) Qualitatively, our learned interest function appears to naturally discover states of importance, such as bottlenecks [38]. Such discovery is demonstrated to be very useful in transfer learning experiments. Our results highlight the general applicability of emphatic algorithms beyond the off-policy single-task setting considered in most previous studies of emphatic RL.

## 2    Background

We assume a Markov Decision Process $\mathcal{M}$, defined as a tuple $\langle \mathcal{S}, \mathcal{A}, r, P \rangle$ with a finite state space $\mathcal{S}$, a finite action space $\mathcal{A}$, a transition probability distribution $P : \mathcal{S} \times \mathcal{A} \times \mathcal{S} \to [0, 1]$, and a scalar reward function $r(s, a)$ depending on action $a \in \mathcal{A}$ in state $s \in \mathcal{S}$. The policy $\pi : \mathcal{A} \times \mathcal{S} \to [0, 1]$ specifies the agent's behaviour and its expected discounted return starting from any state is represented as the value function: $V^\pi(s) = \mathbb{E}_\pi \left[ \sum_{i=t}^\infty \gamma^{i-t} R_{i+1} \big| S_t = s \right]$, where $\gamma \in [0, 1)$ is the discount factor and $R_{t+1}$ is the sampled reward after performing action $A_t$ in state $S_t$. Under linear function approximation, the value function is defined with parameters $\theta \in \mathbb{R}^n$ and features $\phi(s) \in \mathbb{R}^n$, that is $\hat{V}^\pi(s; \theta) = \theta^\top \phi(s)$[2]. An efficient family of algorithms for learning such functions builds on the Temporal Difference (TD) algorithm [39] where the value parameters, $\theta$, are updated as follows:

$$\theta_{t+1} = \theta_t + \alpha(R_{t+1} + \gamma \theta_t^\top \phi_{t+1} - \theta_t^\top \phi_t)\phi_t \tag{1}$$

with $\alpha$ denoting the step size. In the control setting, the policy gradient theorem [45] for the episodic case provides the gradient of the expected discounted return from an initial state distribution $d(s_0)$ with respect to a stochastic policy $\pi(\cdot \mid s; \nu)$ now parameterized by $\nu$:

$$\frac{\partial J_\pi(\nu)}{\partial \nu} = \sum_s d_\pi^\gamma(s) \sum_a \frac{\partial \pi(a|s; \nu)}{\partial \nu} Q^\pi(s, a) \tag{2}$$

where $d_\pi^\gamma(s) = \sum_{s_0} d(s_0) \sum_{t=0}^\infty \gamma^t P^\pi(S_t = s | S_0 = s_0)$ is the discounted state occupancy measure of the target policy $\pi$ and $Q^\pi(s, a) = \mathbb{E}_\pi \left[ \sum_{i=t}^\infty \gamma^{i-t} R_{i+1} \big| S_t = s, A_t = a \right]$ is the state-action value function. For a more detailed presentation of the notation please refer to App. C.

### 2.1    Emphatic Algorithms

Emphatic algorithms [44, 27] provide a way to emphasize and de-emphasize the updates made at each iteration while preserving convergence. Their development was motivated by the challenges that arise under off-policy learning when using function approximation and bootstrapping [50, 41]. In the off-policy setting, a behavior policy $b(a|s)$ generates the data to learn value functions or

---

[2]To simplify notation, we drop $s$ in $\phi$ (i.e. $\phi_t$ instead of $\phi(S_t)$ where $S_t$ is the sampled state at time $t$).

policies evaluated under the target policy $\pi(a|s)$. Emphatic algorithms generalize TD in various ways, however of particular interest to our work is the added flexibility of arbitrarily defining the intrinsic importance of each state through the interest function. In the following we present emphatic algorithms in the general off-policy setting, as learning on-policy is a special case.

**Policy Evaluation:** In its simplest one-step bootstrapping form, the Emphatic Temporal Difference (ETD) update rule for the value parameters $\theta$ takes the following form

$$\theta_{t+1} = \theta_t + \alpha\rho_t F_t(R_{t+1} + \gamma\theta_t^\top \phi_{t+1} - \theta_t^\top \phi_t)\phi_t \tag{3}$$

where $\rho_t = \frac{\pi(A_t|S_t)}{b(A_t|S_t)}$ is the importance sampling ratio at time $t$ and $F_t$, the followon trace, is defined as,

$$F_t = i(S_t) + \gamma\rho_{t-1}i(S_{t-1}) + \gamma^2\rho_{t-1}\rho_{t-2}i(S_{t-2}) + ... = i(S_t) + \gamma\rho_{t-1}F_{t-1}$$

where $i(\cdot) : \mathcal{S} \to \mathbb{R}^+$ is the arbitrary user-defined interest function. The specific form of this trace depends in part on the interest function, but also on how much a state is bootstrapped from by previous states, discounted over time. This specific form is what confers stability and convergence to ETD [57], without introducing the full product of importance ratios used for prior correction [30].

**Control:** In the actor-critic setting, [10] proposed to maximize the excursions objective $J_b(\nu) = \sum_s d_b(s)V^\pi(s)$ where $d_b(s)$ is the stationary distribution of the policy $b$. We explain in App. C the reason why the stationary distribution appears instead of the discounted state occupancy measure of (2). They proposed a way to approximate the policy gradient, where such approximation is only valid in the tabular case. [20] later derived the correct gradient for the more general objective that now includes the state dependent interest function,

$$J_b(\nu) = \sum_s d_b(s)i(s)V^\pi(s) \tag{4}$$

where the correct stochastic gradient update for the policy parameters $\nu$ takes the following form,

$$\nu_{t+1} = \nu_t + \alpha F_t\rho_t\nabla \log \pi(A_t|S_t;\nu_t)Q^\pi(S_t, A_t) \tag{5}$$

Interestingly, the same trace $F_t$ from the off-policy policy evaluation setting appears in the off-policy control setting. Note that $\rho_t$ would be equal to 1 in the on-policy setting where target policy and behavior policy are the same.

## 3 Adaptive Interest

The interest function was designed as a way to emphasize some states more than others and as such can be an efficient way to encode useful inductive bias. Although it may be possible to find an interest function that is effective for a specific and simple case (e.g. when additional knowledge about the task is readily available), it is not convenient to hand-design interest functions that effectively work for complex domains. For this reason, most previous works on emphatic algorithms consider a simple uniform interest over states [44, 21].

Furthermore, we hypothesize that the usefulness of a particular interest function can vary through the learning process itself. This is obvious in the case of a changing environment, for example in continual learning, but is also relevant in the single task setting where the agent's policy or bootstrapping targets vary in a non-stationary manner. In the next section, we further motivate the advantage of an adaptive interest function through the example of a simple chain MDP.

### 3.1 Motivating Example

In Fig. 1, we consider the case of off-policy control in a simple chain MDP made of four non-terminal states. Here our agent uses one-step SARSA [32] to learn the action-value function (Q-value). The agent starts in state $S_0$ and can reach either the terminal state on the right with reward of 1 or the more distant terminal state on the left with reward of 100. In our example, suppose the behavior policy is biased towards going right in the three rightmost states. The resulting target policy learned via SARSA without any emphatic weighting is misguided toward a suboptimal solution (Fig. 1). App. B describes additional details of this experiment.

When designing a fixed interest function for SARSA with emphatic weighting, it would be advantageous to emphasize the states on the left and de-emphasize the states on the right, as a way to try and avoid the sub-optimal solution. However, Fig. 1 shows that although a well-designed fixed interest function can improve upon the this baseline, the resulting emphatic SARSA is still unable to converge to a good policy within 500 updates.

Finally we consider using an adaptive interest function. Here we leverage the same pattern as in the previous fixed interest function, but we only activate the interest in some states at certain times. Particularly, at the start of training, only the left-most state is emphasized, and all other states have interest set to near-zero. As credit is propagated from the terminal left state towards the rest of the chain, the interest of the second left-most state is increased. This continues until credit assignment reaches the starting state and the optimal action is selected. Emphatic SARSA with such a dynamic interest function is able to quickly converge to a good policy (Fig. 1).

Fig. 1 empirically shows that, in our example MDP, a standard SARSA agent (without emphatic weighting) is outperformed by emphatic SARSA with a fixed interest (supporting the general utility of emphatic algorithms), which is in turn outperformed by emphatic SARSA with an adaptive interest (supporting the additional utility of adaptive interest). This example highlights that even in tabular settings without function approximation, the additional flexibility of an adaptive interest function can be quite beneficial. It is important to note that the fixed interest agent eventually finds the right solution in our example, but it has much worse sample complexity compared to our adaptive interest agent.

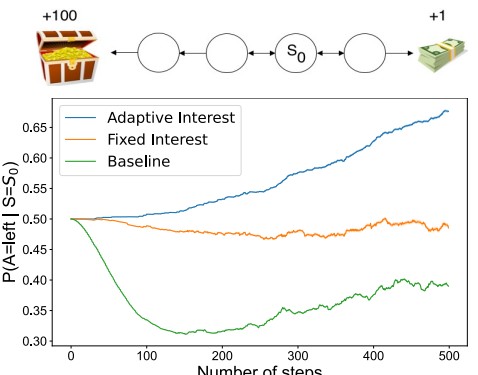

Figure 1: **Four State Chain MDP with off-policy control** where the behavior policy is biased towards going right. We plot the probability that the learned target policy (implemented as a Boltzmann policy) goes left in the initial state $S_0$, which is the optimal action to take. A **baseline** SARSA agent not using emphatic weighting will struggle to overcome this bias, whereas an emphatic agent with **fixed** interest ($[10, 1, .1, .001]$ for each state from left to right) will require many samples to obtain the optimal policy. Using an **adaptive** interest inside emphatic updates guides credit assignment towards the starting state efficiently, converging towards the optimal policy early on. Results are averaged over 500 runs.

### 3.2 Meta-Gradient Interest (MINT)

As discussed in the previous section, we seek an adaptive interest function that would improve the learning process during training. However, automatically discovering such a function is not straightforward, in part as it can take any arbitrary value, and because we have to evaluate the effect of a particular change in the interest function with regards to the agent's parameters. We explore adapting the interest function with a wide variety of heuristics such as the prediction error and find that such approaches do not generally provide improvements. Alternatively, meta-gradients [55, 63] are a natural candidate as they can automatically discover such interest functions at each stage of learning through the interaction of an inner loop and an outer loop of optimization. We therefore propose to learn and adapt the interest function parameterized by meta-parameters $\eta$ in an online manner, within a single lifetime and within a single environment.

During the inner loop, the meta-parameters $\eta$, together with the agent's policy and value parameters $\{\nu, \theta\}$, appear in the base objective, $J^B(\theta, \nu, \eta)$. Only the parameters are updated through this objective while the meta-parameters $\eta$ remain fixed and influence the gradients. To illustrate the influence of the meta-parameters on the resulting parameters, we can write them as functions of $\eta$, i.e. $\{\nu'(\eta), \theta'(\eta)\}$.

During the outer loop, the updated parameters are evaluated with respect to a meta-objective, $J^M(\nu'(\eta), \theta'(\eta))$, from which we derive the gradients with respect to $\eta$. This is referred to as the *meta-gradient*, which evaluates how the values of the meta-parameters affected the performance of the updated parameters. By repeating this process, meta-gradients will adapt the meta-parameters in order to more efficiently improve the parameters themselves.

We now describe the specific choices behind applying meta-gradients to emphatic algorithms when updating the policy. For simplicity, the derivation for the value function is relegated to App. H.3.

In the inner loop, the agent maximizes the following inner objective,

$$J^B(\nu, \eta) = \sum_s d_b(s) i(s; \eta) V^\pi(s) \qquad (6)$$

where the interest function $i$ is parameterized by the meta-parameters $\eta$ and the policy $\pi$ is parameterized by $\nu$. This inner objective is based on the excursions objective [20], that is, the future reward achieved by following the target policy $\pi$ starting from the distribution of states generated by the behavior $b$. Another possibility would have been to consider the counterfactual objective [59] or the alternative life objective [29], however these choices imply additional complexities which we leave for future work. We provide a more detailed discussion on the choice of the objective function in the App. H.1. The inner loop can be written by using (5), obtaining:

$$\nu' \leftarrow \nu + \alpha_b \rho_t F_{t,\eta} \nabla_\nu \pi(A_t | S_t; \nu) Q^\pi(S_t, A_t) \qquad (7)$$

where $F_{t,\eta} = i(S_t; \eta) + \gamma \rho_{t-1} F_{t-1,\eta}$ emphasizes the current state according to the current meta interest and the followon trace at the previous timestep.

When considering the meta-objective, practitioners usually employ the same form as the inner objective. In our case this would be written as

$$J^M(\nu'(\eta)) = \sum_s d_b(s) V^\pi(s) \qquad (8)$$

where $\pi$ is defined by the updated parameters $\nu'(\eta)$. Recently, [14] argue that such an approach may lead to a poor meta-optimisation landscape, as both objectives share the same curvature. In App. D, we verify different meta-objectives, such as the variance of the reward-to-go [47], and report no increase in performance when compared to (8) in our setting. It is likely that obtaining their improvements also relies tackling myopia in meta-gradients. From (8) we obtain the following meta-gradient

$$\eta' \leftarrow \eta + \alpha_m \nabla_\eta J^M = \eta + \alpha_m \nabla_{\nu'} J^M \nabla_\eta \nu' \qquad (9)$$

where $\nabla_\eta \nu'$ encodes how the meta-parameters affected the new parameters. A stochastic sample of this quantity at time $t$ can be expanded as (see also App. H.2),

$$\left( \sum_{i=0}^t \gamma^{t-i} \nabla_\eta i_\eta(S_i) \rho_{i:t} \right) \nabla_\nu \log \pi(A_t | S_t; \nu) Q^\pi(S_t, A_t)$$

where $\rho_{i:t}$ is a product of importance sampling ratios. Pseudocode for our approach, which we call **MINT** (**M**eta-gradient **Int**erest), is presented in Algorithm 1 of App. A.

Performing the update rule in (9) would require a new set of samples. In practice, the same samples are re-used for both loops [63, 62, 51] through a sliding window of experience. In our work we opt to use the importance sampling ratio method of [63]. Finally in the present derivation we only consider 1-step meta-gradient [52] as it greatly simplifies the exposition ( see App. H.4 for derivations). We also show in App. H.4 why the sampling correction term [2] does not appear in the off-policy setting.

## 4 Experiments

We now validate our method on a wide range of scenarios to assess the following questions: 1) Can we automatically learn an interest function on complex environment in order to improve performance? 2) How robust is the meta interest with respect to the agent's hyperparameters? 3) Is it possible to leverage the information encoded by the learned interest function for downstream tasks?

We first conduct experiments in the off-policy policy evaluation setting under linear function approximation. Next, we extend the usual field of study of emphatic algorithms and verify their general utility. In particular, we study how they can improve the performance of on-policy algorithms under the control setting, where considerable gains are witnessed only under an adaptive interest. Finally, we further extend our investigation to the transfer learning setting by leveraging the interest function learned in a previous task in order to greatly speed up the learning process in a second task. All hyperparameters are available in the App. E.

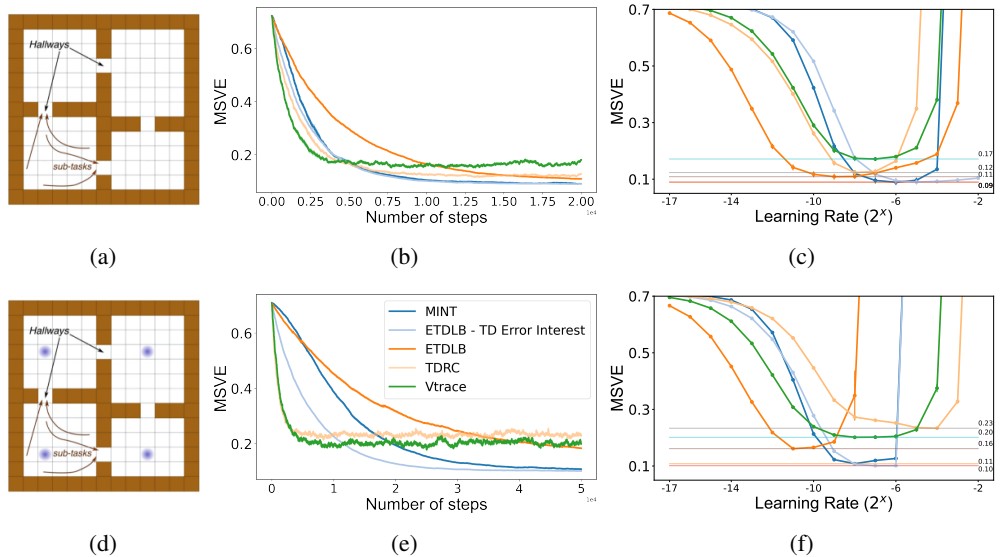

(a)            (b)            (c)

(d)            (e)            (f)

Figure 2: **Off-Policy Evaluation under linear function approximation** where we build on the empirical setup of [16]. The top row presents results for the 4Rooms-8Tasks (4R8T) domain, while the bottom row presents for the HighVariance-4Rooms-8Tasks (H4R8T) domain, where states in blue present high variance. We compare emphatic algorithms to high performing off-policy baselines such as TDRC and Vtrace. For both environments we notice that by adapting the interest function, either through meta gradients as in MINT or through the absolute value of TD error, the final prediction error is significantly improved, especially in the second environment where MINT *reduces by half* the error when compared to non-emphatic methods.

## 4.1 Linear Function Approximation

**Setup.** Our first experiments are done in the off-policy policy evaluation setting with linear function approximation using tile coding [41]. We adopt the setup from [16], who considered two variations of the classical Four Rooms domain, shown in Fig. 2. We name these variations 4Rooms-8Tasks (4R8T) and HighVariance-4Rooms-8Tasks (H4R8T) to highlight their characteristics (See App. D for their details).

**Quantitative Results.** Fig. 2 shows the results for the 4R8T (top) and H4R8T (bottom row). The y-axis indicates the mean squared value error (MSVE) averaged across all policies. In Fig. 2c and 2f, we vary the learning rate on the x-axis and for each value report each algorithm's best final performance chosen across all values of the bootstrapping coefficient $\lambda$ (and other possible hyperparameters). Fig. 2b and 2e show the best learning curves for each algorithm.

In these two domains, we compare learning the interest through meta gradients, MINT, to a baseline that adapts the interest with respect to the absolute value of the TD error (ETDLB - TD Error Interest, where ETDLB refers to the generalized version of emphatic TD [19]). We compare these adaptive methods to the standard emphatic baseline ETDLB. Finally, we also compare to the recent TD with Regularized Corrections (TDRC) algorithm [15], which follows the line of work on Gradient TD [43], as well as the V-trace [11], which is representative of the performance of methods that use truncated importance sampling ratios [26]. We notice that across both tasks as shown in Fig. 2e, leveraging an adaptive interest leads to a better final value error and especially on H4R8T, our method *almost halves the error* when compared to non-emphatic methods.

In App. D we present additional figures that take into account a different metric: the area under the curve (instead of the final performance) and notice a similar pattern. When looking at the learning curves on the left, we notice that ETDLB pays a price in terms of slower convergence in order to achieve a better final performance. However, when using an adaptive interest, the difference with non-emphatic methods is greatly reduced, especially in the 4R8T domain.

Examining closely Fig. 2c and 2f, we notice that leveraging an adaptive interest moves the bottom of the U-shaped curve to the right. By selectively emphasizing some states, emphatic algorithms using an adaptive interest are able to learn on a higher learning rate. However, the shape of the U curve tends to cut drastically after a certain threshold, at which point the updates become unstable.

When we compare adapting the interest through meta gradients to the one defined as the absolute value of the TD error, we notice that their performance is almost equal. It can perhaps seem surprising that this would be case, as the meta gradients method can in theory learn any function. For our particular choice of objectives, we show in App. H.3 the form of the meta gradients under linear function approximation. For simplicity, if we further assume that features are tabular and we are under the on-policy setting, we get that the stochastic sample at time $t$ of the gradient is,

$$\nabla_\eta J_t^M = \mathbf{e}_t(\delta_t)^2 \tag{10}$$

where $\mathbf{e}_t$ indicates the one-hot vector represent the current state at time $t$. (10) shows that the meta gradient updates in the direction of the squared TD error. Although the convergence of the meta-parameters will not be to the sampled squared TD error, at each iteration the meta-parameters are affected in a similar way when compared to the absolute value of the TD error heuristic. In practice, the features are not exactly tabular and therefore the updates made on one state may affect another, which would also explain why MINT is slower to converge.

**Qualitative Results.** In Fig. 3, we inspect the learned interest functions obtained by MINT on four of the eight tasks (one per room). The top row shows the learning process for 4R8T, while the bottom row shows the process for H4R8T. In both domains, the first state to be highlighted is the one next to the goal in the hallway. Indeed this state is highly influential since all states bootstrap from it, directly or indirectly.

As training progresses, the interest in the 4R8T diffuses to neighbouring states (this bears a close resemblance to the example presented in Fig. 1 ). In the H4R8T, a different pattern emerges where the state with high variance is being highlighted early on in training. As this state is visited by many trajectories that the target policy would take, it influences the values of many other states that need to bootstrap from it. However, since it exhibits higher variance than neighbouring states, it requires more computation to be correctly estimated.



(a) 4Rooms-8Tasks (4R8T)

(b) HighVariance-4Rooms-8Tasks (H4R8T)

Figure 3: **Visualization of the interest function across iterations.** These show results at the start (left column), mid-training (middle column) and at the the end (right column). Depending on the environment, different patterns are being encoded in the interest function. In the H4R8T environment, the high variant state is being emphasized as it requires more resources in order to be estimated accurately.

At the end of training, we observe that some states are particularly less important, like state near the opposite corners of the hallways. As the target policy does not visit them often and not many states bootstrap from them, it is less important for them to be accurate. Moreover, it is interesting to note that the diffusion of interest observed in the 4R8T domain is not perfectly uniform. Since tile coding [41] is used as the function approximator, a specific pattern of the interest function may be needed at different states to lower the overall prediction error.

## 4.2 Experiments at Scale

Emphatic algorithms were initially derived for the off-policy setting. However, the flexibility given by the interest function is generally applicable, even in the on-policy case. To showcase this flexibility, in this section we investigate the performance of emphatic algorithms in the on-policy setting using non-linear function approximation.

### 4.2.1 MinAtar

**Setup.** We verify the generality of the proposed method by considering the MinAtar domain [56], which is a miniaturized version of some of the games from the classic Atari 2600 testbed [6]. The environment provides $10 \times 10 \times n$ state representations, where $n$ varies for each game. The environments are implemented using sticky actions and randomization [24]. For all games we use 10 random seeds and report the mean and standard deviation after 10M timesteps.

**Results.** Fig. 4 shows that MINT provides good gains when compared to the two baselines. A standard PPO agent [37] and a meta-gradient approach [54] that meta-learns the target function which appears in RL update rules. We explain in detail this baseline in the App. F and theoretically show

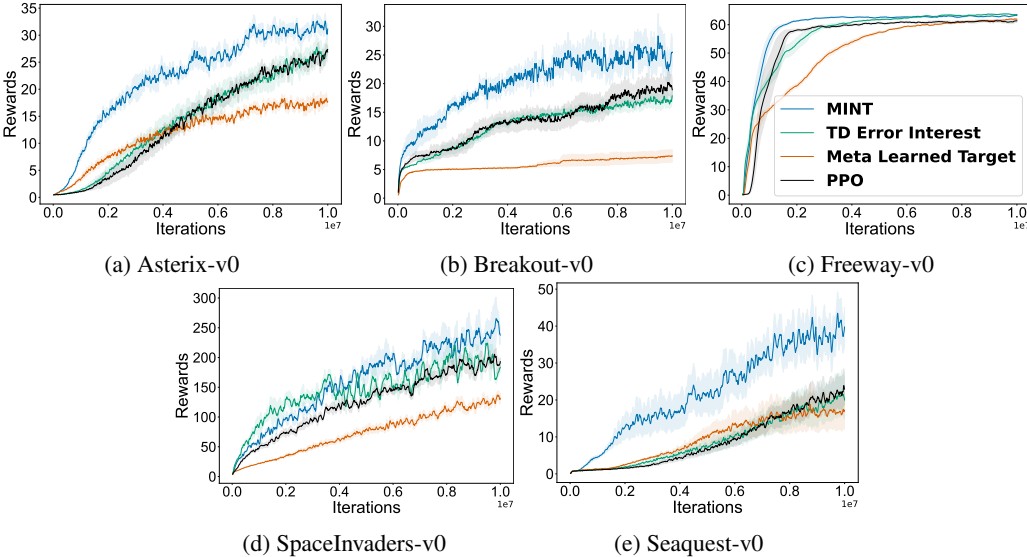

Figure 4: **Performance on MinAtar.** Adapting the interest provides consistent gains in sample efficiency as well as final performance. Meta-learning the target function is a more general approach than ours, but also requires more samples to provide improvements over the standard baseline.

that learning the interest function is not simply special case of their approach. Results in Fig 4 clearly demonstrate that our method outperforms both baselines. Our findings also agree with the experiment in [54] (Fig 3a in their paper) that meta-learning the target function requires many more samples before it can match the baseline's performance. [54] also compares to an approach that meta-learns the complete loss function (which could recover our update rule) and find that the agent is not able to learn in the online, single-lifetime setting (which is our setting).

These results highlight the difficulties of more general meta-learning formulations and their impact on sample efficiency. This suggests that meta-learning the interest-function may be a good trade-off between generality and the amount of inductive bias. We additionally present results in Fig. 10 where we vary the learning rate and present at the U shaped curves of performance, which seem to behave similarly to the linear function approximation case.

### 4.2.2 Continuous Control

**Setup.** We perform experiments on the MuJoCo domain [49, 7], where states and actions are continuous. We report the mean and standard error averaged across 10 random seeds. We include several emphatic baselines where we explore using a fixed interest and various heuristics for adapting the interest function. Additionally, we investigate the usefulness of adapting the learning rate itself using hypergradient descent [5]. We provide a description of all the baselines in App. E.

**Results.** As Fig. 5 shows, utilizing adaptive interest function is the key to get consistent improvement over PPO (across almost all environments). Interestingly, the adaptive heuristic based on the TD error that worked well in the prediction setting does not generalize to this one. One way to understand this is to consider that a low TD error may not be indicative of a high performing policy in control.

We also compare our method to hypergradient descent (HD) [5] which dynamically updates the learning rate during training. We notice that HD does not seem to provide gains, except in Humanoid-v3 where it reaches the performance of MINT at the cost of a slower learning process. This is in contrast to our method which generally does not suffer from increased sample complexity to achieve better performance. We highlight that an important difference between HD and MINT is that the interest function is a state-dependent quantity, which can provide additional flexibility. In the App. E we compare to additional baselines, such as meta-learning the reward function [63] and various adaptive heuristics for the interest function.

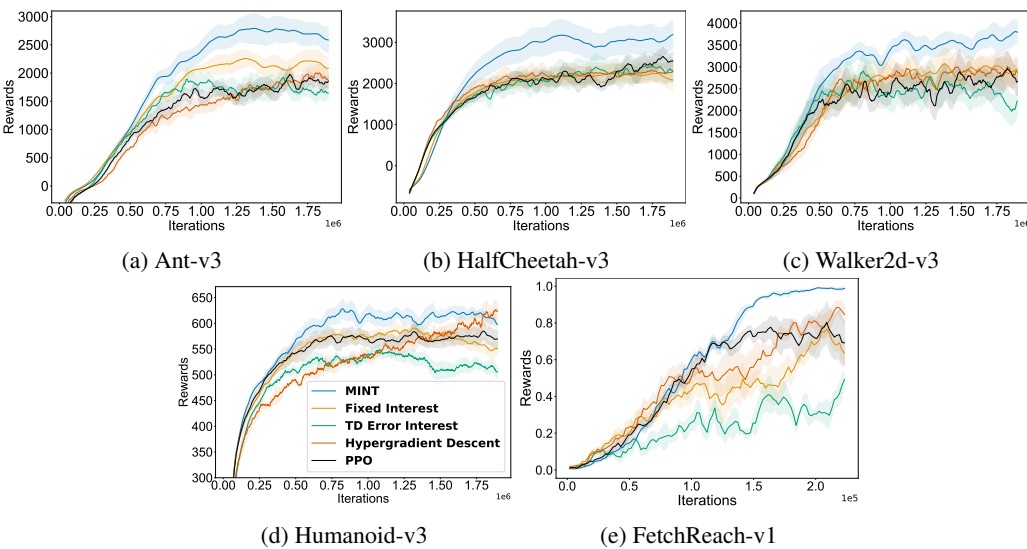

(a) Ant-v3       (b) HalfCheetah-v3       (c) Walker2d-v3

(d) Humanoid-v3       (e) FetchReach-v1

Figure 5: **Results on continuous control.** We compare MINT to various baselines including an emphatic variant of PPO using a fixed interest, as well as an interest based on the absolute value of the TD error. We also verify whether updating the learning rate via hypergradient descent can match the performance of MINT. Across environments, we notice that *adapting the interest via meta-gradients* is key to obtain consistent gains.

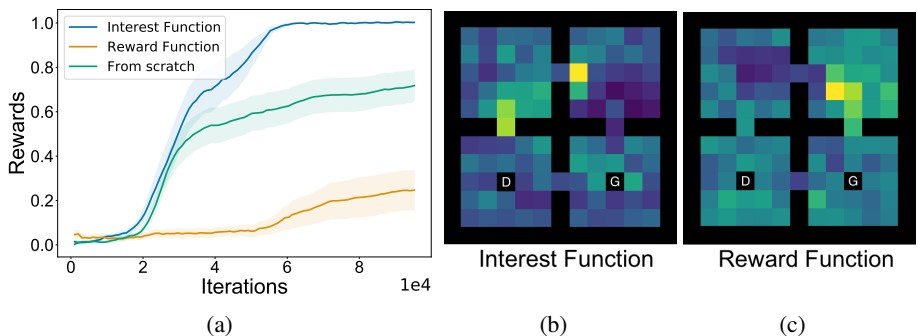

(a)       (b)       (c)

Figure 6: **Transfer experiment** where the interest or the reward function is transferred to help an agent learn a policy from scratch in a variant of the task. In b) and c) we visualize the values of each function, where brighter color means higher value. Interestingly, the interest function highlights states near the hallways of the starting state room (top-left), also referred to as *bottleneck states*.

## 4.3 Transfer Learning across RL Tasks

**Setup.** As the interest function is automatically learned, we observe that it likely encodes knowledge and information that can be useful later. In particular, this knowledge may serve to speed up learning in new environment over learning from scratch. We investigate this hypothesis in the FourRoomsTransfer environment [9], shown in Fig. 6a where we show mean and standard deviation across 30 seeds. The agent starts in the top left corner and has to get to the goal location in green. The state highlighted in orange is a distractor state that provides a random reward, $R \sim \mathcal{N}(0, 1)$.

For the transfer setting, we change the location of each of these entities (see App. G). In this setting, the agent learns in the first environment and only transfers the interest function to the second one, which remains fixed thereafter. We also compare to an agent that uses meta gradients to learn an intrinsic reward function before transferring it to the second task, similarly to [62].

**Results.** Fig. 6 shows that the learned interest function provides a significant speed-up when compared to the actor-critic baseline that learns from scratch without an interest function.

Here, transferring the intrinsic reward does not seem to help. Note we *do not* claim that using meta-gradients to learn an intrinsic function is always a better choice than to learn an intrinsic reward function, as it likely depends on the exact transfer learning setup[3].

To understand how the interest function helps in transfer, we present in Fig. 6b and Fig. 6c the learned interest function and the learned reward function. We notice that the interest function highlights states that are near the goal, but also the hallways of the starting room (top left), which are usually referred to as bottleneck states [38]. Such states are of particular importance as they influence the trajectory an agent takes as well as many of the predictions it makes during such trajectory. As we notice, the interest function highlights the hallway leading to the goal, but also the one leading to the distractor. On the opposite, the reward function naturally highlights the path to the goal and de-emphasizes the one leading to the distractor state. This illustrates one useful property of the interest function: it highlights the location of rewards, whether they are positive or negative. This kind of invariance is key for a better transfer performance in our setting, and could be used more generally in continual learning [31]. This experiment also points to an interesting future direction where a universal interest function could be defined similarly to universal value functions [33].

## 5 Related Work

**Emphatic Methods**. Initially derived by [44, 27] as a stable and simple one-time-step solution to the problem of off-policy prediction. Its convergence is shown in [57] when employing the full trace and later [61] show convergence for the Truncated ETD algorithm. The ideas in prediction were extended to control by [20, 18, 60]. Emphatic algorithms have been shown to be a strong baseline in many benchmarks under linear function approximation [16, 17], even in the on-policy case [53, 3]. Recently, emphatic algorithms have been extended to the deep RL by building on a variant of the IMPALA agent [11] with auxiliary heads and have shown superior performance on Atari [21, 22].

**Selective Updating**. Emphatic methods can be seen more generally as performing selective updating, whereby through a scalar we emphasize or de-emphasize the updates to each state. This has previously been investigated in the context of model free learning [34] and model based planning [1]. Recently, [8] provide a unifying view of various algorithms as a form of selective credit assignment. In hierarchical reinforcement learning, selective updating is an integral part of the options framework either through initiation sets [46] or interest functions [23].

**Meta Gradients.** Meta learning [35, 48, 13] is a class of methods that have better capabilities in adapting to new tasks by learning a prior from previously seen related tasks in the past [36, 4]. While these methods mainly focus on multi-task learning [12], meta-gradients based methods [62, 63] focus on learning the meta parameters online within a single task, based on online cross validation [40].

## 6 Conclusion

We propose to learn and adapt the interest function utilizing meta-gradients in an online fashion where hand-coded solutions are not feasible. Comprehensive experiments on various settings suggest that automatically adapting the interest function from a stream of data leads to improved performance. Although certain heuristics for adapting the interest function are occasionally beneficial, our experiments point that consistency and general usefulness are achieved through our adaptive method.

---

[3]Note, we do not consider using recurrent neural networks or a lifetime value function here, which were found critical for meta-learned intrinsic rewards to work well in transfer [62]. Also, the interest/reward functions are trained on a single environment at first, before being transferred to a new one, without further training.

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
