# Appendix: Adaptive Interest for Emphatic Reinforcement Learning

## A  Algorithm

---
**Algorithm 1** MINT
---
1: Initialize policy and value parameters $\{\nu, \theta\}$
2: Initialize meta-parameters for the interest function $\{\eta\}$
3: **while** not done **do**
4:     Generate trajectory either using the target policy (on-policy case) or the behavior policy (off-policy case).
5:     Update policy and value function using Equations 7-16
6:     Update interest function using Equation 9.
7: **end while**

---

## B  Motivating Example Implementation Details

In the example of Figure 1, the behavior policy chooses to go right with probability 0.9, which misguides the agent towards a sub-optimal policy as it will visit the right terminal state considerably more often. Notice that the left terminal state gives a higher reward. We consider using SARSA [32] to learn the agent's action value function. To obtain a probability distribution over actions, we use a Boltzmann policy in conjunction of the action value function with unitary temperature. As we see in Figure 1 where we plot the probability of going left in the starting state $S_0$, the baseline SARSA agent not using emphatic is stuck with a sub-optimal policy after 500 steps. We additionally consider using emphatic weightings with a fixed interest function. The specific values of this function are $[10, 1, 0.1, 0.001]$ for each of the four states in the chain MDP, going left to right. Notice that the values are increasing as we go left, as an attempt to counter the behavior policy's bias towards going right. This agent performs much better than the baseline, however it may require many more samples to finally converge to the optimal solution. The agent using an adaptive interest function is on the other hand much more sample efficient. This agent also uses the values $[10, 1, 0.1, 0.001]$ for the interest function, however it only activates some values at certain stages of learning. In particular, at first, only the left most value is active, and the remainder is set to near-zero. Once credit assignment has reached the left most state, the first and second values are active, while the third and fourth remain masked. As more credit assignment is propagated, more values of the interest become active. With this simple strategy we notice in Figure 1 that the agent using an adaptive interest is prefers to go left in the starting state at almost anytime during learning.

## C  Background and Notation

In the main text we present the TD and ETD algorithms for policy evaluation under linear function approximation, as a way to recognize the existing literature on emphatic algorithms [27]. We here present the derivation for policy evaluation under general function approximation. Following standard notation [41], capital letters for states, actions or rewards represent the random variable at time $t$ (i.e. $S_t$ is the random variable at time $t$) and lowercase letters represent their instantiation (i.e. $S_t = s$ is the random variable $S_t$ taking value $s$ at time $t$). The TD algorithm under general function approximation updates the value function parameters $\theta$ in the following way,

$$\theta_{t+1} = \theta_t + \alpha\big(R_{t+1} + \gamma\hat{V}^\pi(S_{t+1}; \theta_t) - \hat{V}^\pi(S_t; \theta_t)\big)\nabla_{\theta_t}\hat{V}^\pi(S_t; \theta_t)$$

where $\hat{V}^\pi$ is the function approximation to the true value function $V^\pi$ when following policy $\pi$. The true value function is defined as the expected discounted return from a given state when following policy $\pi$,

$$V^\pi(s) = \mathbb{E}_\pi\big[\sum_{i=t}^\infty \gamma^{i-t}R_{i+1}|S_t = s\big]$$

We can also define the action value function as the expected discounted return from a given state and given action when following policy $\pi$,

$$Q^\pi(s,a) = \mathbb{E}_\pi\Big[\sum_{i=t}^\infty \gamma^{i-t}R_{i+1}|S_t = s, A_t = a\Big] = \mathbb{E}[R_{t+1} + \gamma V^\pi(S_{t+1})|S_t = s, A_t = a]$$

In the control setting, we use function approximation to parameterize the policy $\pi(a|s;\nu)$ using parameters $\nu$. In the episodic on-policy setting, we update the policy using gradient ascent on the following objective,

$$J_\pi(\nu) = \sum_s d_0(s)V^\pi(s)$$

where $d_0$ is an arbitrary starting state distribution. Using the policy gradient theorem [45], we get the following,

$$\frac{\partial J_\pi(\nu)}{\partial \nu} = \sum_s d_\pi^\gamma(s) \sum_a \frac{\partial \pi(a|s;\nu)}{\partial \nu} Q^\pi(s,a)$$

where $d_\pi^\gamma(s) = \sum_{s_0} d(s_0) \sum_{t=0}^\infty \gamma^t P^\pi(S_t = s|S_0 = s_0)$ is the discounted state occupancy measure of the policy $\pi$. The quantity $P^\pi(S_t = s|S_0 = s_0)$ is defined as

$$P^\pi(S_t = s|S_0 = s_0) = \prod_{i=1}^{t-1} \sum_{S_i} \sum_{A_i} P(S_{i+1}|S_i, A_i)\pi(A_i|S_i;\nu) \sum_{A_0} P(S_1|S_0, A_0)\pi(A_0|S_0;\nu)$$

using the environment transition distribution $P$ and the policy $\pi$.

Policy evaluation done under general function approximation using ETD algorithm to update the value function parameters takes the following form,

$$\theta_{t+1} = \theta_t + \alpha F_t \rho_t \big(R_{t+1} + \gamma \hat{V}^\pi(S_{t+1};\theta_t) - \hat{V}^\pi(S_t;\theta_t)\big)\nabla_{\theta_t}\hat{V}^\pi(S_t;\theta_t)$$

where $F_t$, the followon trace, is defined as,

$$F_t = i(S_t) + \gamma\rho_{t-1}i(S_{t-1}) + \gamma^2\rho_{t-1}\rho_{t-2}i(S_{t-2}) + ... = i(S_t) + \gamma\rho_{t-1}F_{t-1}$$

The emphatic algorithm for off-policy control is defined through the excursions objective generalized through the state-dependent interest function,

$$J_b(\nu) = \sum_s d_b(s)i(s)V^\pi(s)$$

where $d_b$ is the stationary distribution of the behavior policy $b$. The excursions objective is based on the continuing setting and as a result the starting state distribution $d_0$ is replaced with the behavior stationary distribution $d_b$. Intuitively, this objective encodes the expected return from executing the target policy starting from the distribution of states. It is possible to reconcile the continuing setting, which is more theoretical in nature, to the episodic setting, which is more practical, by considering generalizations of the discount factor and bootstrapping parameter [78].

## D    Linear Function Approximation

We borrow the experimental setup from [16] which considers the off-policy prediction setting. The agent uses linear function approximation to learn a value function for a target near-optimal policy while the data is generated by an near-uniform policy. We also adopt the environmental setup from them which they considered two variations of the classical Four Rooms domain (depicted in Figure 2). We name these variations 4Rooms-8Tasks and HighVariance-4Rooms-8Tasks to highlight their characteristics.

In both domains, the agent starts randomly in any state and follows an infinitely long trajectory where the value functions are updated online. Each of the four rooms consists of two independent tasks: reaching each of the two hallways. As such, a total of eight value functions for eight target policies are being evaluated simultaneously, where each of the policies follows the shortest path to the hallway location.

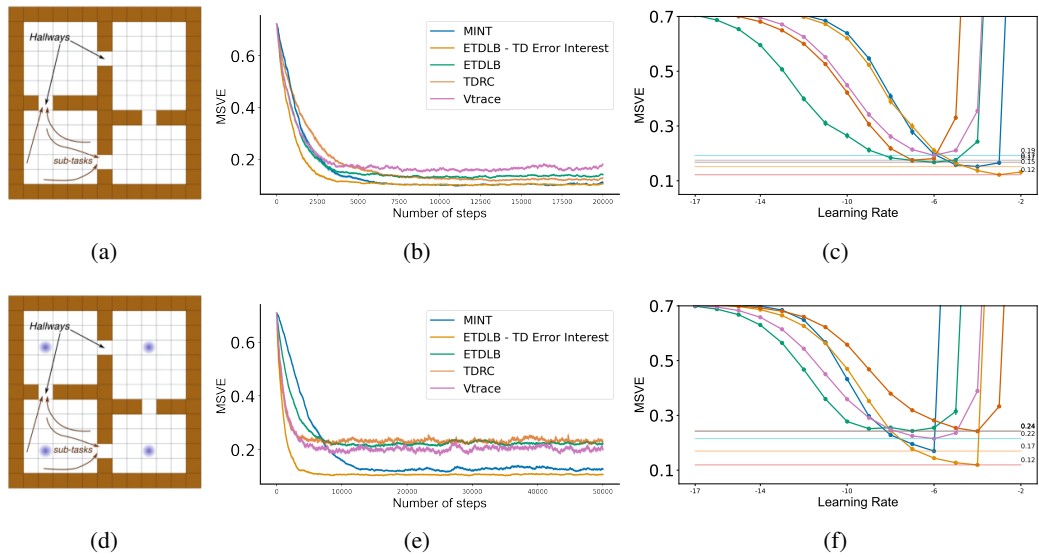

(a)                        (b)                                        (c)

(d)                        (e)                                        (f)

Figure 7: **Off-Policy Evaluation under linear function approximation** where we build on the empirical setup of [16]. The top row presents results for the 4Rooms-8Tasks domain, while the bottom row presents for the HighVariance-4Rooms-8Tasks domain, where states in blue present high variance. We compare emphatic algorithms to high performing off-policy baselines such as TDRC and Vtrace. For both environments we notice that by adapting the interest function, either through meta gradients as in MINT or through the TD error, the area under the curve is significantly improved.

In 4Rooms-8Tasks, the behavior policy is simply a uniform distribution over actions. In the HighVariance-4Rooms-8Tasks domain, the behavior policy is uniform, except for the states highlighted in blue, as shown in Figure 2d, where it takes one particular action with probability $0.97$: in two left rooms this action is to go left, whereas in the two right rooms it is to go right.

The setup from [16] benchmarks a wide variety of algorithms, from which we choose some of the most performing ones. We also pick these algorithms such that they are representative of different families of algorithms. In Figure 2 we presented the results in terms of final performance. We present in 7 the results in terms of area under curve (AUC). We notice once again that adapting the interest function can provide improvements, especially on HighVariance-4Rooms-8Tasks environment. The performance of MINT compared to the ETDLB baseline using the absolute value of the TD error is slightly lower in terms of AUC. Meta-optimization can bring a set of challenges that need to be addressed in order to improve on the performance we report. In particular we did not use any special tricks such as gradient clipping.

For each algorithm, we plot the best performing curves by looking across learning rate values $2^{-x}$ where $x \in \{0, 1, 2, .., 18\}$ and bootstrapping coefficient $\{0.0, 0.1, 0.2, 0.3, 0.5, 0.9, 1.0\}$. For the emphatic algorithms we searched the $\beta$ parameter in $\{0.0, 0.2, 0.4, 0.6, 0.8, 1.0\}$. We used a meta learning rate of $0.25$ for both environments and across all other values of hyperparameters. The interest function is implemented using tabular representation to allow for better stability of the meta-optimization. We do not use any activation function for the interest function as it naturally remains greater than zero.

We also try a couple of different meta objectives, as a way to decouple the curvature of the meta optimisation from the optimisation of the base objective. We present in Fig. 8 results for the reward to go (RTG) [76] and variance temporal difference learning (VTD) [79]. We do not observe an improvement in performance. This highlights that the meta optimisation landscape is complex in nature and researchers would benefits by probing further how to improve the curvature, as suggested by [14].

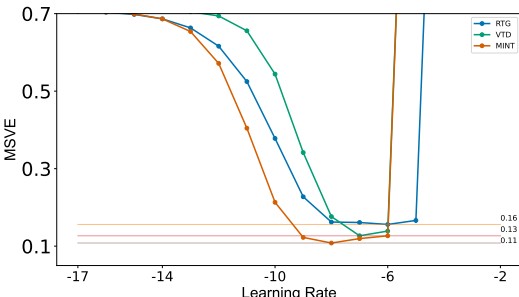

Figure 8: Off-Policy Evaluation under linear function approximation where we build on the empirical setup of [16]. We compare MINT using the excursions objective as the meta-objective to MINT using the reward to go (RTG) and variance temporal difference learning (VTD) as part of the meta-optimisation.

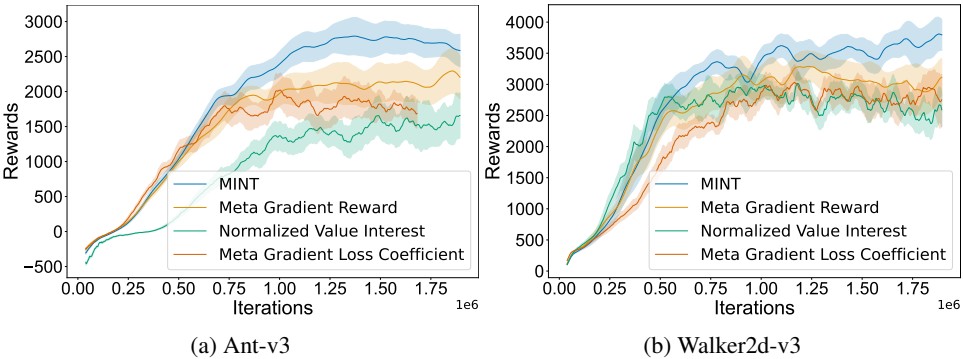

|               |               |
|:-------------:|:-------------:|
| (a) Ant-v3    | (b) Walker2d-v3 |

Figure 9: **Additional Baselines on MuJoCo.** We compare MINT to various baselines including an emphatic variant of PPO using the sigmoid of the value function. Additionally we use meta-gradients to learn the loss coefficient as in [58]. Finally, we also try learning an intrinsic reward funtcion through meta gradients, as a way to verify whether the same compute resources used for learning the interest function can be better spent.

# E   Continuous Control

We consider the standard environment from [7] and compare our approach to various baselines building on top of PPO [37]. In Figure 5 we consider an emphatic version of PPO where the interest is fixed (**Fixed Interest**) and where the interest is defined through the absolute value of the TD error (**TD Error Interest**). Additionally, we compare with a method using hypergradient descent [5] (**Hypergradient Descent**) to learn the learning rate.

In Figure 9 we provide another set of baselines. We verify adapting the interest function by setting it to the sigmoid of the value function (**Normalized Value Interest**), with the intuition that rewarding states may be more valuable. However, this does not help. We also used meta gradients to learn the loss coefficient (**Meta Gradient Loss Coefficient**) as in [58]. This method was shown to provide significant improvements, at the condition of using auxiliary losses, which we do not employ. Finally, we verify whether using the resources for learning an interest function can be better served by learning other quantities. In particular, we try learning an intrinsic reward function (**Meta Gradient Reward**), which helps more than other baselines but still does not match the performance of our method.

Across all environments we use the default hyperparameters from [66] for the base learner. For the meta learning rate we verify values in $\{1 \times 10^{-4}, 3 \times 10^{-4}\}$, and we use the former for all games, except Walker2d-v3 where the latter provided slightly better performance. The neural network for the interest function is the same network used for the policy and value function: a two layer MLP with hidden sizes $\{64, 64\}$. The interest function uses an exponential activation.

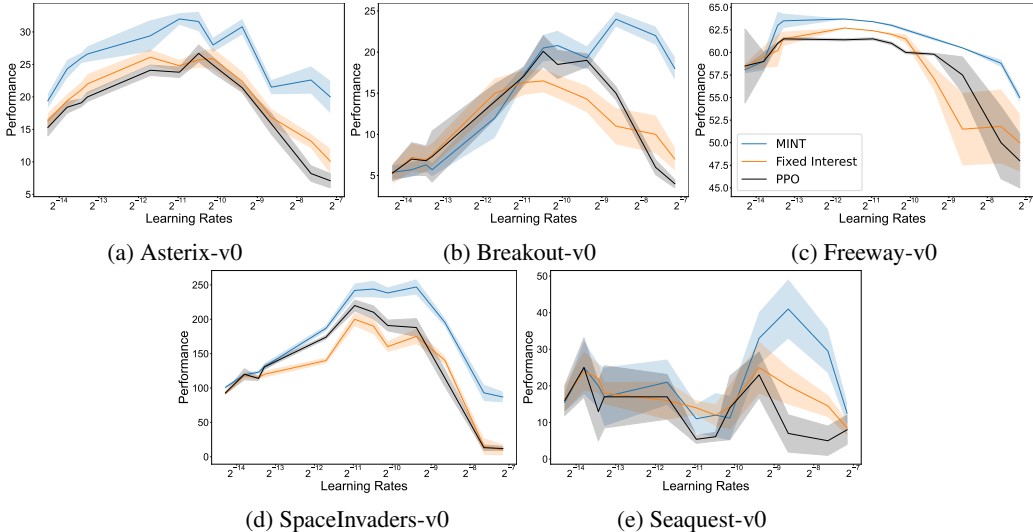

Figure 10: **Sensitivity analysis on MinAtar.** Across almost all learning rates, adapting the interest is key to improve upon the baseline. For most of the games, MINT's inverse-U shaped curve tends reaches its peak for *higher learning rates* than the baseline.

# F  MinAtar

For the experiments on MinAtar we use convolutional neural networks for feature extraction within the interest function, the policy and value function. Across all environments and across all learning rates we use the same default meta learning rate of $1 \times 10^{-4}$. The interest function uses an exponential activation.

In [54], the authors propose to apply meta-gradients to meta learn the target function $G(s)$ used to update the policy or the value function. This target function appears in the following update rule. First it appears when updating the value function's parameters $\theta$,

$$\theta' = \theta + \alpha(G(S_t) - \hat{V}^\pi(S_t;\theta))\nabla_\theta \hat{V}^\pi(S_t;\theta) \tag{11}$$

It also appears when considering actor critic algorithms and updating the policy,

$$\nu' = \nu + \alpha\nabla_\nu \pi(A_t|S_t;\nu_t)(G(S_t) - \hat{V}^\pi(S_t;\theta)) \tag{12}$$

The target function in RL can be a simple one step bootstrapping target $R_{t+1} + \gamma\hat{V}^\pi_\theta(s_{t+1})$, to more complex mixes of n-step returns parameterized by the bootstrapping parameter $\lambda$ [42]. In [54] this is instead replaced by a parameterized function $G_\psi(s)$ where $\psi$ are the meta parameters. In the inner loop of optimization, the agent updates its value parameters through the following rule,

$$\theta' = \theta + \alpha(G_\psi(S_t) - \hat{V}^\pi(S_t;\theta))\nabla_\theta \hat{V}^\pi(S_t;\theta) \tag{13}$$

We can contrast this equation with our update rule,

$$\theta' \leftarrow \theta + \alpha_b F_t \rho_t (R_{t+1} + \gamma V^\pi(S_{t+1};\theta) - V^\pi(S_t;\theta))\nabla_\theta V^\pi(S_t;\theta) \tag{14}$$

$$\theta' \leftarrow \theta + \alpha_b \Big(\sum_{i=0}^{t} \gamma^{t-i} i(S_i;\eta)\rho_{i:t}\Big)\rho_t (R_{t+1} + \gamma V^\pi(S_{t+1};\theta) - V^\pi(S_t;\theta))\nabla_\theta V^\pi(S_t;\theta) \tag{15}$$

It is possible to see by inspection that the effect of the interest function is on the whole update rule, that is, it is multiplicative. In contrast, the target function is only subtracted from the current estimate of the value function. Therefore, meta-learning the interest function can not be reduced to a special case of learning the target function as they play two different roles in the update rule. A similar argument can be made for the the policy updates.

# G MiniGrid Transfer

For the experiments on MiniGrid-FourRoomsTransfer-v0 [9], we also use convolutional neural networks for feature extraction within the interest function, the policy and value function. In the MiniGrid-FourRoomsTransfer-v0 environment, the agent must reach the goal location within 60 steps. There is also a distractor state that outputs a random reward distributed as a Gaussian with mean zero. We use a learning rate of $3 \times 10^{-4}$. The interest function uses an exponential activation. For the transfer learning setting, we consider the following setup,

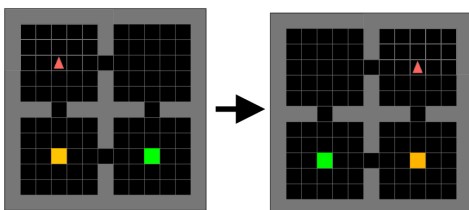

Figure 11: **Transfer learning setting.** We first train the agent on the environment on the left, before vary the location of each entity in the environment. In this new environment we transfer the previously learned interest function or reward function.

# H Meta-Gradients

## H.1 Choice of objective

The meta-objective is based on the excursions objective, that is,

$$J_b(\nu) = \sum_s d_b(s)i(s)V^\pi(s)$$

Another possibility would have been to consider the alternative life objective,

$$J_\pi(\nu) = \sum_s d_\pi(s)i(s)V^\pi(s)$$

The latter better encodes the performance of the target policy after deployment, as the states are weighted according to the target policy $\pi$ instead of the behavior policy $b$. However, optimising this objective will require estimating the density ratio between the stationary distribution of the behavior and target policies, that is $\frac{d_\pi(s)}{d_b(s)}$. Although methods exist [68], they usually require more complex optimisation methods and have not yet shown great performance on complex domains. Another possibility for the policy evaluation case is to consider at each timestep $t$ the full ratio of importance sampling ratios between the behavior and target policies [29, 30]. However, this approach is prone to excessively large variance.

Another possibility is to consider mixing the excursions and alternative life objective through a scalar $\hat{\gamma} \in [0, 1]$ to obtain the counterfactual objective [59], which is written as,

$$J_{\hat{\gamma}}(\nu) = \sum_s d_{\hat{\gamma}}(s)i(s)V^\pi(s)$$

where $d_{\hat{\gamma}}(s)$ is defined as,

$$\mathbf{d}_{\hat{\gamma}} = (1 - \hat{\gamma})(\mathbf{I} - \hat{\gamma}\mathbf{P}_\pi^\top)\mathbf{d}_b$$

When $\hat{\gamma} = 1$, we recover the alternative life objective, whereas when $\hat{\gamma} = 0$ we recover the excursions objective. Optimising this objective requires keeping track of two traces. In practice, it was shown to provide improvements on control tasks. As the best performing agent had a value of $\hat{\gamma}$ was closer to the excursions objective, we decide to proceed with the standard excursions objective to avoid the additional traces.

## H.2 Policy parameters

Consider the base learner objective

$$J^B = \sum_s d_b(s) i(s; \eta) V^\pi(s)$$

with the associated base learner updates,

$$\nu' \leftarrow \nu + \alpha_b F_t \rho_t \nabla_\nu \log \pi(A_t | S_t; \nu) Q^\pi(S_t, A_t)$$

For the meta-objective we consider the following excursions objective,

$$J^M = \sum_s d_b(s) V^\pi(s)$$

The meta updates can then be written as,

$$\eta' \leftarrow \eta + \alpha_m \nabla_\eta J^M$$

where using the chain rule gives us,

$$\nabla_\eta J^M = \nabla_{\nu'} J^M \nabla_\eta \nu'$$

Expanding the second term, we get that the stochastic sample of the gradient is,

$$
\begin{aligned}
\nabla_\eta \nu' &= \nabla_\eta \big(\nu + \alpha_b F_t \rho_t \nabla_\nu \log \pi(A_t | S_t; \nu) Q^\pi(S_t, A_t)\big) \\
&= \nabla_\eta \big(\alpha_b F_t \rho_t \nabla_\nu \log \pi(A_t | S_t; \nu) Q^\pi(S_t, A_t)\big) \\
&= \nabla_\eta \big(\alpha_b \sum_{i=0}^t \gamma^{t-i} i(S_i; \eta) \rho_{i:t} \nabla_\nu \log \pi(A_t | S_t; \nu) Q^\pi(S_t, A_t)\big) \\
&= \alpha_b \sum_{i=0}^t \gamma^{t-i} \nabla_\eta i(S_i; \eta) \rho_{i:t} \nabla_\nu \log \pi(A_t | S_t; \nu) Q^\pi(S_t, A_t)
\end{aligned}
$$

where we used the log derivative trick [45] and the definition of the followon trace.

## H.3 Value parameters

Consider the base learner objective,

$$J^B = \sum_s d_b(s) i(s; \eta) \frac{1}{2} (v^\pi(s) - V^\pi(s; \theta))^2$$

where $v^\pi(s)$ is the true value function (which is inaccessible), with the following base learner updates,

$$\theta' \leftarrow \theta + \alpha_b F_t \rho_t (R_{t+1} + \gamma V^\pi(S_{t+1}; \theta) - V^\pi(S_t; \theta)) \nabla_\theta V^\pi(S_t; \theta) \qquad (16)$$

where we consider the semi-gradient update rule [41]. Notice that different values of the bootstrapping coefficient will lead to different update rules [44]. Consider the following meta-objective,

$$J^M = \sum_s d_b(s) \frac{1}{2} (v^\pi(s) - V^\pi(s; \theta'(\eta)))^2$$

with the following meta-update,

$$\eta' \leftarrow \eta + \alpha_m \nabla_\eta J^M$$

where

$$\nabla_\eta J^M = \nabla_{\theta'} J^M \nabla_\eta \theta'$$

In practice the term $\nabla_{\theta'} J^M$ will as well be a semi-gradient due to bootstrapping. In this equation, $\nabla_\eta J^M$ is of size $1 \times m$, where $m$ is the size of the meta-parameters $\eta$. Consequently, $\nabla_{\theta'} J^M$ is of size $1 \times n$, where $n$ is the size of the parameters $\theta$ and $\nabla_\eta \theta'$ is of size $n \times m$.

Consider the case where the parameters and meta parameters use linear function approximation, that is $i(s; \eta) = \eta^\top \psi(s)$ and $V^\pi(s; \theta) = \theta^\top \phi(s)$, where $\phi$ are the features for the value function and $\psi$ are the features for the interest function. Furthermore, consider the on-policy case. We can then write that the stochastic sample at time $t$ is,

$$\nabla_{\theta'} J_t^M = \phi_t \delta_t$$

where $\delta_t$ is the TD(1) error and $\phi_t$ indicates the features at time $t$. We also have that

$$\nabla_\eta \theta_t' = D_{\phi_t} \Psi_t \delta_t$$

where $D_{\phi_t}$ is the diagonal matrix with $\phi_t$ on its diagonal and where $\Psi_t$ is the matrix where each row is made of the interest function's features $\psi_t$.

If we further consider that all features are tabular representations, we have that,

$$\nabla_{\theta'} J_t^M = \mathbf{e}_t \delta_t$$

where $\mathbf{e}_t$ is the one-hot vector that indicates the state at time $t$. We also have that $\nabla_\eta \theta'$ is a matrix where all elements are zero, except at position $s, s$ (the state at time $t$) where it is equal to $\delta_t$. Multiplying the $\nabla_{\theta'} J^M$ and $\nabla_\eta \theta'$ together gives us,

$$\nabla_\eta J_t^M = \mathbf{e}_t (\delta_t)^2$$

That is, the meta gradient updates in the direction of the $L_2$ norm of the TD error.

## H.4  Multi-step derivation

In the main text we present the derivation for the meta-gradient obtained from one step updates in the inner loop. This was done to help the clarity of the presentation, but in practice it is possible to do multiple updates to the parameters before updating the meta-parameters. This sequence of K updates can be written as,

$$\nu_K = \nu_0 + \sum_{k=0}^{K-1} U(\eta, \nu_k, \mathcal{D}_k)$$

where $U$ is the update rule of (7) using the data $\mathcal{D}_k$ collected at iteration $k$ (i.e. a set of trajectories). If the data is collected on-policy, taking the gradient of meta-objective with respect to the meta parameters would result in,

$$\nabla_\eta J^M(\eta) = \sum_{k=0}^{K} \mathbb{E}_{\{\mathcal{D}_i\}_{i=0}^{k-1}} \mathbb{E}_\pi \Big[ \sum_t \Big( \sum_{j=0}^{k-1} \nabla_\eta \nu_j \nabla_{\nu_j} \log p(\mathcal{D}_j | \nu_j) +$$
$$\nabla_\eta \nu_k \nabla_{\nu_k} \log \pi(A_t | S_t; \nu_k) \Big) Q^\pi(S_t, A_t) \Big]$$

Let's explain this equation. The sum over $k$ comes from considering the evaluation of the meta-objective after each of the inner updates. This is known to optimize the area under the curve of the meta-objective after each update, rather than final performance [77]. The outer expectation is over the datasets collected at each of the previous iterations, as this data affects the current parameters. The sum over $t$ is with respect to the current trajectory obtained from the current parameters. The second term multiplying the action value function, $\nabla_\eta \nu_k \nabla_{\nu_k} \log \pi(A_t | S_t; \nu_k)$ is the part of the meta gradient that considers the effect of the meta parameters on the current trajectory's performance.

The first term multiplying the action value function, $\sum_{j=0}^{k-1} \nabla_\eta \nu_j \nabla_{\nu_j} \log p(\mathcal{D}_j | \nu_j)$, is the *sampling correction* term and was first derived by [2]. As the current parameters were obtained by updates using sampled datasets, the parameters are random variables as well. The sampling correction term then allows for the meta gradients to propagate through inner update iterations. It is usually not included in most of the meta gradient literature, which renders the meta updates biased. However, including them may lead to increased variance [52], and as such there is a trade-off to consider.

In the off-policy control case using the excursions objective, the sampling correction term does not appear as the data is collected through a behavior policy, which is not influenced by the meta-parameters. However, under the alternative life objective we would have extra terms similar to the sampling correction as optimising this objective would require using the full product of importance

sampling ratios between target and behavior policies, or the ratio between stationary distributions of the target and behavior policies. Please see section App. H.1 for more details on the alternative life objective.

In the policy evaluation setting, whether we are on or off-policy, the sampling correction term would also not appear, as the parameters of the policy generating the data are not updated. In the policy evaluation setting only the value function estimating the expected return is parametrised.

## Additional References for the Appendix

[64] A. G. Baydin, R. Cornish, D. M. Rubio, M. Schmidt, and F. Wood. Online learning rate adaptation with hypergradient descent. In *International Conference on Learning Representations*, 2018.

[65] G. Brockman, V. Cheung, L. Pettersson, J. Schneider, J. Schulman, J. Tang, and W. Zaremba. Openai gym. *CoRR*, abs/1606.01540, 2016.

[66] P. Dhariwal, C. Hesse, O. Klimov, A. Nichol, M. Plappert, A. Radford, J. Schulman, S. Sidor, Y. Wu, and P. Zhokhov. Openai baselines. https://github.com/openai/baselines, 2017.

[67] S. Ghiassian and R. S. Sutton. An empirical comparison of off-policy prediction learning algorithms in the four rooms environment. *CoRR*, abs/2109.05110, 2021.

[68] Q. Liu, L. Li, Z. Tang, and D. Zhou. Breaking the curse of horizon: Infinite-horizon off-policy estimation. *CoRR*, abs/1810.12429, 2018.

[69] A. R. Mahmood, H. Yu, M. White, and R. S. Sutton. Emphatic temporal-difference learning. *CoRR*, abs/1507.01569, 2015.

[70] A. Patterson, A. White, S. Ghiassian, and M. White. A generalized projected bellman error for off-policy value estimation in reinforcement learning. *CoRR*, abs/2104.13844, 2021.

[71] D. Precup, R. Sutton, and S. Dasgupta. Off-policy temporal-difference learning with function approximation. *Proceedings of the 18th International Conference on Machine Learning*, 06 2001.

[72] J. Schulman, F. Wolski, P. Dhariwal, A. Radford, and O. Klimov. Proximal policy optimization algorithms. *arXiv preprint arXiv:1707.06347*, 2017.

[73] R. S. Sutton and A. G. Barto. *Reinforcement Learning: An Introduction*. The MIT Press, second edition, 2018.

[74] R. S. Sutton, A. R. Mahmood, and M. White. An emphatic approach to the problem of off-policy temporal-difference learning. *J. Mach. Learn. Res.*, 17(1):2603–2631, jan 2016.

[75] R. S. Sutton, D. McAllester, S. Singh, and Y. Mansour. Policy gradient methods for reinforcement learning with function approximation. In *Advances in Neural Information Processing Systems*, volume 12. MIT Press, 2000.

[76] A. Tamar, D. D. Castro, and S. Mannor. Learning the variance of the reward-to-go. *Journal of Machine Learning Research*, 17(13):1–36, 2016.

[77] V. Veeriah, M. Hessel, Z. Xu, R. L. Lewis, J. Rajendran, J. Oh, H. van Hasselt, D. Silver, and S. Singh. Discovery of useful questions as auxiliary tasks. *CoRR*, abs/1909.04607, 2019.

[78] M. White. Unifying task specification in reinforcement learning. *CoRR*, abs/1609.01995, 2016.

[79] M. White and A. White. A greedy approach to adapting the trace parameter for temporal difference learning. *CoRR*, abs/1607.00446, 2016.

[80] S. Zhang, W. Boehmer, and S. Whiteson. Generalized off-policy actor-critic. In *Advances in Neural Information Processing Systems*, volume 32, 2019.