# OpenReview forum: "Adaptive Interest for Emphatic Reinforcement Learning"
_NeurIPS.cc/2022/Conference — NeurIPS 2022 Accept_

### Official Review · Reviewer_2sr2 · 2022-06-15

**Rating:** 6
**Confidence:** 5
**Soundness:** 4 excellent
**Presentation:** 4 excellent
**Contribution:** 3 good

**Summary:**

This paper proposes to use meta-gradient to adapt the interest function in emphatic RL algorithms. Empirical study covering off-policy policy evaluation with linear function approximation, on-policy control with nonlinear function approximation, as well as transfer learning confirm the efficacy of the proposed method.

**Questions:**

See "Weaknesses"

**Strengths And Weaknesses:**

Strength:
1. This paper is well motivated and targets an important problem in RL. The use of meta-gradient is straightforward but appears novel to me.
2. The paper is well organized and easy to follow. Empirical study covers a wide rage of settings.

Weaknesses:
1. The jump from off-policy prediction to on-policy control seems a bit abrupt to me. Emphatic methods are originally designed for off-policy prediction so starting with Section 4.1 seems natural to me. But is there any particular reason that the control experiments are in the on-policy setting, especially given the success of [22, 23], the motivating example in Section 3.1, and the derivation in Section 3.2. I feel adding more off-policy control experiments would significantly strengthen this work.
2. The meta-objective in Equation (8) seems not well motivated. Why not simply use the episodic return \sum_s p_0(s) v_\pi(s) where p_0 denotes the initial distribution? After all all the domains used for control is episodic instead of continuing and the agents are evaluated against episodic return. Is the ease of optimization the main consideration for the use of the excursion objective? I feel more discussion about this would also strengthen this work.

---

> ### Author Response · Authors · 2022-08-02
> **Response to Reviewer 2sr2**
>
> Thank you for your valuable feedback. We hope you will consider increasing your score after reading our responses. Please let us know if there are more questions.
>
> > The jump from off-policy prediction to on-policy control seems a bit abrupt to me. Emphatic methods are originally designed for off-policy prediction so starting with Section 4.1 seems natural to me. But is there any particular reason that the control experiments are in the on-policy setting, especially given the success of [22, 23], the motivating example in Section 3.1, and the derivation in Section 3.2. I feel adding more off-policy control experiments would significantly strengthen this work.
>
> We agree that adding another off-policy experiment could help the current work, however we believe that we currently probe a variety of settings and that it might be difficult to make everything fit inside a paper. Performing off-policy control in the function approximation setting with neural networks comes with complications. The big hurdle is that off-policy often relies on importance sampling ratios and it is not straightforward to use them efficiently with current neural networks and methods/algorithms surrounding them. As this creates confounding factors, we decided to focus on the simpler on-policy case in order to obtain a clearer picture of the effectiveness of our approach.
>
> In particular, the work in [22] would be the most promising avenue, however it is important to highlight that the setting considered in section 4 of their paper is not the standard off-policy setting. In particular, in their paper, the agent still acts on-policy with respect to the target policy (the main head). The auxiliary heads, which learn off-policy, are not evaluated in terms of how accurate their predictions are (or in terms of executing their policies and observing the discounted return), but instead are considered as a way to improve the representation/feature space for the main head. We do not try to diminish their results, but simply to highlight an important difference. Future work considering head-on the off-policy setting with neural networks would have to carefully choose the setting and the kind of choices needed.
>
>
> > The meta-objective in Equation (8) seems not well motivated. Why not simply use the episodic return \sum_s p_0(s) v_\pi(s) where p_0 denotes the initial distribution? After all all the domains used for control is episodic instead of continuing and the agents are evaluated against episodic return. Is the ease of optimization the main consideration for the use of the excursion objective? I feel more discussion about this would also strengthen this work.
>
> Thank you for pointing this out. The main reason for considering the excursion objective is for ease of mathematical derivation of the update rule. The continuing setting can encompass the episodic setting as discussed in White, 2017 by considering a transition-based discount factor, this way the problem specification can be independent of the way the MDP is implemented. We will make sure to add this information to the main paper.
>
>
> [White, 2017]: White, Martha. ​​Unifying Task Specification in Reinforcement Learning, 2017.

---

### Official Review · Reviewer_UBwo · 2022-07-09

**Rating:** 7
**Confidence:** 2
**Soundness:** 3 good
**Presentation:** 4 excellent
**Contribution:** 3 good

**Summary:**

This paper extends work focusing on emphatic TD learning. While the original formulation of emphatic TD learning included an interest function that prioritises updates to certain states, in practice this was always made uniform or required expert knowledge to set. The idea here is to (meta) learn the interest function as training progresses - an inner loop follows regular emphatic TD learning with the learned interest function, while an outer loop next runs to update the interest function's parameters. Empirical results are provided in several domains, including pixel-based ones, continuous control tasks and linear function approximation settings. Results indicate that an adaptive interest function learned over time seems key to applying these approaches to on-policy learning, while the method also outperforms a fixed interest function, as well as other meta-parameter learning approaches.


**Questions:**

1. In the MinAtar experiments, I was expecting to see the TD-error heuristic which was used in the chain walk (and popped up again in MuJoCo). Is there a reason that it wasn't included in this set of experiments?

2. On Line 292 it states: "We explain in detail this baseline in the App. F and theoretically show that learning the interest function is not simply (sic) special case of their approach". However, I cannot find a discussion of this anywhere.

3. In the motivating example, I'm not entirely clear on why the choice of base learning algorithm was SARSA. Is this because the focus, later on, is on-policy learning? Otherwise, it seems strange to apply the various interest strategies on top of a learning algorithm that doesn't quite fit.

4. Is there any intuition for why the MSVE should diverge for larger learning rates with the TD error interest on H4R8T, but not 4R8T?

As an aside (certainly not necessary to include here), the focus on the robustness of hyperparameters like learning rate could potentially benefit from applying the methodology of [1].

Minor comments:

1. It would be helpful to move Figure 4 to the top of the next page. Currently, it appears before Fig 3 and you have to jump between pages to read the associated text.
2. Without knowledge of the exact reward function, Figure 3 takes some puzzling out. My guess is that it is sparse (1 at the goal, 0 elsewhere) but that is simply a guess because I do not see it listed anywhere

3. Many of the references list the arxiv or preprint versions of papers. Please check these all, since many have been published. For one example: [59] appeared at NeurIPS 2019.

Minor typos:
L184: "as THE inner objective"
L239: algorithm -> algorithms

[1] Jordan, Scott, et al. "Evaluating the performance of reinforcement learning algorithms." International Conference on Machine Learning. PMLR, 2020.

**Strengths And Weaknesses:**

\+ I found the paper to be extremely well-written and well-motivated. Every time a sentence raised a question in my mind, the very next sentence explained or justified it to me!

\+ The main idea is conceptually simple (in a good way) and the paper does a good job of evaluating it in several different kinds of domains.

\- I realise that space is a large issue here, but I found that I had to jump back and forth between the appendix and the main paper quite often. While much of this is extra discussion, a few times it was for key information, such as the details of the domain. In particular, I'm still not 100% certain if the reward function for the four rooms domain is a sparse one (and what the exact values are), even having consulted the appendix.

---

> ### Author Response · Authors · 2022-08-02
> **Response to Reviewer UBwo**
>
> Thank you for your valuable feedback. Please let us know if there are more questions.
>
> > In the MinAtar experiments, I was expecting to see the TD-error heuristic which was used in the chain walk (and popped up again in MuJoCo). Is there a reason that it wasn't included in this set of experiments?
>
> Thank you for your suggestion, we ran the new experiments and have added it to the paper (​​Figure 4). Here is a convenient link for the results: https://imgur.com/pP1ZzJT . These results show that, as in the MuJoCo experiments, the TD error interest baseline does not perform well in control, which is in contrast to the results in policy evaluation. This also highlights that meta-gradients are more flexible and more adaptable to the specific setting.
>
> > On Line 292 it states: "We explain in detail this baseline in the App. F and theoretically show that learning the interest function is not simply (sic) special case of their approach". However, I cannot find a discussion of this anywhere.
>
> Good catch and thank you. It seems that somehow we missed this and forgot to include it. We have now updated the Appendix F with the derivation.
>
> > In the motivating example, I'm not entirely clear on why the choice of base learning algorithm was SARSA. Is this because the focus, later on, is on-policy learning? Otherwise, it seems strange to apply the various interest strategies on top of a learning algorithm that doesn't quite fit.
>
> We opted for SARSA as it is a value based control algorithm that can conveniently be derived in an emphatic version. An alternative would have been to consider an actor-critic algorithm, however this would have created an interplay between the approximate critic and the actor which adds complexity to the toy example and creates confounding factors. We use SARSA in an off-policy manner by making use of importance sampling ratios to leverage data generated by a behavior policy. We will make sure to add this clarification to the paper.
>
> > Is there any intuition for why the MSVE should diverge for larger learning rates with the TD error interest on H4R8T, but not 4R8T?
>
> In the H4R8T domain, there is one state in particular for each room (highlighted in blue in Figure 2d) that exhibits high variance, which in general destabilizes the learning process. If by chance when the agent visits that state the TD error is high, and the importance sampling ratio is very large (the variance), the combination of the two with a large learning rate might push the parameters too far in a certain unpromising direction, from which the agent could not recover. In 4R8T, the importance sampling ratios remain pretty small throughout the state space and as such the learning process is more stable. That being said, this is an intuitive description of what is happening and most likely we would need to confirm it by more experiments that could uncover interesting phenomena.
>
>
> > As an aside (certainly not necessary to include here), the focus on the robustness of hyperparameters like learning rate could potentially benefit from applying the methodology of [1].
>
> We were not aware of this paper but will definitely go through it and if possible adopt their methodology to report some of our results.
>
> > It would be helpful to move Figure 4 to the top of the next page. Currently, it appears before Fig 3 and you have to jump between pages to read the associated text.
>
> Good suggestion, thanks. We have updated the paper to address this.
>
> > Without knowledge of the exact reward function, Figure 3 takes some puzzling out. My guess is that it is sparse (1 at the goal, 0 elsewhere) but that is simply a guess because I do not see it listed anywhere
>
> Yes that is right, we will add this important detail in the main paper.
>
> > Many of the references list the arxiv or preprint versions of papers. Please check these all, since many have been published. For one example: [59] appeared at NeurIPS 2019.
>
> Thank you for pointing this out, we have now updated the paper to address this.

---

> > ### Comment · Reviewer_UBwo · 2022-08-08
> > **Response**
> >
> > Thanks to the authors for answering my questions and clarifying matters. I've read through the other reviews and comments. It feels like the paper could do with some more detail and discussion in various places, but I still think that the paper makes a good contribution to the literature and successfully demonstrates the advantage of learning the interest function from data.

---

> > > ### Author Response · Authors · 2022-08-09
> > > **Thanks**
> > >
> > > Thanks again for suggesting those experiments, comments, and responses. Please let us know if you have more questions or comments.

---

### Official Review · Reviewer_xj8y · 2022-07-11

**Rating:** 6
**Confidence:** 4
**Soundness:** 3 good
**Presentation:** 3 good
**Contribution:** 3 good

**Summary:**

This paper studies the problem of learning the interest function, which emphasizes the importance of different states differently in RL objectives. The interests of different states are typically set to be all 1 in existing approaches. And this paper seems to pioneer the study of the adaptive interest function (the paper didn't claim it but from its discussion of existing works it appears to me that the authors believe that no other works have studied this problem before).

The paper proposed a meta-learning approach to the problem. Specifically, the interest function is learned to maximize an objective, which is an off-policy control objective. The paper proposed an off-policy prediction algorithm and an on-policy control algorithm. Empirically, the off-policy prediction algorithm almost matches the performance of an ETD algorithm with TD error as the interest. The on-policy control algorithm is significantly better than the PPO algorithm and the other Meta-learning algorithm.

**Questions:**

Overall, could you explain how different interests influence the learning process in prediction and control settings respectively?

line 21 A possible solution could be to selectively emphasize certain updates, for example through a state-dependent interest function.
Why is this a possible solution?

Are there any existing methods that adaptively choose the interest? Is your work the first one studying the adaptive interest function?

It is not clear from the introduction section the prediction or the control setting or both is considered in the paper.

When designing a fixed interest function for SARSA with emphatic weighting, it would be advantageous to emphasize the states on the left and de-emphasize the states on the right, as a way to try and avoid the sub-optimal solution.
Why?

Can interests go arbitrarily large?

In figure 1, no algorithm eventually converges to the optimal policy (left), right? Why is that?

In figure 2, how did you compute the MSVE?

Isn't figure 2 showing that the proposed method is slightly worse than the ETDLB method with the TD error as the interest in the tested problem (c.f. subfigure c)?

I didn't find in the paper a description of the off-policy policy evaluation algorithm (the one you used to produce figure 2).

It seems that the interest function is shared across all sub-tasks? If so, why is it reasonably to share it? Why do you consider the multi-task setting instead of the single one?

Is your off-policy policy evaluation algorithm a batch algorithm? Are TDRC, Vtrace, and ETDLB online algorithms or batch algorithms?

In figure 3, I see that during the entire training period, cells near the hallway cells have high interests. I can also see that, at the end of training, the further a state is from from the hallway state, the lower interest the state is assigned. The paper explains that "as the target policy does not visit them often and not many states bootstrap from them, it is less important for them to be accurate". This is a reasonable explanation. But given that the pattern looks pretty close to the value function, it can also be hypothesized that higher state value results in higher interest. If a state has a high value but is not frequently visited by the target policy, would the interest be high?

What is the on-policy control algorithm?

When comparing with PPO and the Meta Learned Target algorithm, do you use the same amount of buffer to store past experience and the same amount of computation?

**Limitations:**

Overall I would need the questions listed above to be addressed to tell the paper's limitations. The paper does not have a potential negative societal impact.

**Strengths And Weaknesses:**

Strengths:
In general, the paper is easy to follow.
Adapting the interest function seems to be interesting and important, and is not well understood in the literature.
The proposed algorithm uses an elegant idea.
Empirical results show some interesting patterns, which would be helpful for understanding the proposed algorithm.

Weakness:
The motivation of the idea, the description of the algorithm, and the explanation of the empirical results are not clear enough. Some important questions need to be addressed to evaluate the paper's contribution. See the Questions section for details.

-------------------------------------------------------------------------------

I have read the authors' responses and feel that most of my questions were answered, including the key question about explaining how interest function influences the learning process. Therefore I decided to increase my score to 6.

---

> ### Author Response · Authors · 2022-08-02
> **Response to Reviewer xj8y - Part 1**
>
> Thank you for your valuable feedback. We hope you will consider increasing your score after reading our responses. Please let us know if there are more questions.
>
> > Overall, could you explain how different interests influence the learning process in prediction and control settings respectively?
>
> There may be a few intricacies to finding a good answer to this question, but we can perhaps start with how the interest function can equally influence both prediction and control. Going back to the original emphatic TD paper (Mahmood et al. 2015), the interest function was introduced as a way to control how the agent spends its computational/representational budget. Not all states can be perfectly estimated, and therefore the accuracy of our estimates in a subset of states will be more crucial in order to obtain a better policy or better predictions.
> In prediction, the interest function only influences where we put more computational resources. This is also true in control, however since the interest affects the policy, it would indirectly also affect the data collection process (in the on-policy case).
>
> As an example, in the prediction case, a natural heuristic is the TD error, as a high error in a state might indicate we have something important to learn from it. This does not translate necessarily to the control case (at least not when considering an actor-critic setting) as high TD errors do not mean that the policy is choosing the wrong actions, but that the critic is not perfect. In the control case, we only care that the policy achieves a high return, in which case we should consider heuristics that reflect this aspect.
>
> > line 21 A possible solution could be to selectively emphasize certain updates, for example through a state-dependent interest function. Why is this a possible solution?
>
> The interest function is a possible solution as it can allocate the computational capacity of the agent towards certain states, potentially towards states that can be most beneficial for improving its performance. It is defined as $i(s): \mathcal{S} \xrightarrow[]{} \mathbb{R}^+$. This function is arbitrary, it can be anything that the user decides to, but the intent is that we can choose which states are important.  To give an example, if we use non-emphatic RL, the agent will update the states that it visits the most. However, it is possible that a state that does appear rarely in the trajectories is actually key to finding a better solution. The interest function could highlight each visit to that state and therefore the agent would more accurately estimate its value. When this value reaches a certain threshold the agent would prefer taking the more optimal path going through the state of interest. The challenge remains in how to identify such states of interest.
>
>
> > Are there any existing methods that adaptively choose the interest? Is your work the first one studying the adaptive interest function?
>
> While previous works such as Zhang et al. 2019 briefly discussed the importance of interest functions which are non-trivial (i.e. to do so, they define $i(s) = \hat{i}(s) c(s)$ where $c(s)$ is density ratio and $\hat{i}$ is a user-defined interest function), to the best of our knowledge, our paper is the first work that comprehensively studies importance of adapting the interest function in empathic RL and proposes to use meta-gradient to learn and adapt interest function.
>
> > It is not clear from the introduction section the prediction or the control setting or both is considered in the paper.
>
> Thank you for pointing this out, we will update the presentation of the introduction to make it clear.
>
> > When designing a fixed interest function for SARSA with emphatic weighting, it would be advantageous to emphasize the states on the left and de-emphasize the states on the right, as a way to try and avoid the sub-optimal solution. Why?
>
> Good question. The reason for this is because the behavior policy that generates the data has a preference for going right, to the sub-optimal terminal state. The non-emphatic agent would update all the states it sees from the experience generated from the behavior policy and would learn very quickly to prefer this sub optimal state. It might take many samples for the agent to first experience the optimal reward (on the left), and then to learn that it is actually the optimal reward and adapt its action value function. By emphasizing states on the left, the rare visits to this optimal reward are strongly highlighted and the visits to the sub-optimal state are tuned down, giving a better chance to the agent to learn the right move.

---

> > ### Author Response · Authors · 2022-08-02
> > **Response to Reviewer xj8y - Part 2**
> >
> > > Can interests go arbitrarily large?
> >
> > By definition, the interest is unbounded, as can be seen in the original emphatic TD paper (Mahmood et al., 2015) and in our presentation of section 2.1. In practice, we haven’t seen the interest go towards infinity, but instead it stabilizes around certain values and behaves well. We will add plots to the appendix that show its behavior across iterations. That being said, it is not impossible that it would go arbitrarily large in the off-policy case as it depends on importance sampling ratios that can grow arbitrarily large. However, we have never seen that happen in practice.
> >
> >
> > > In figure 1, no algorithm eventually converges to the optimal policy (left), right? Why is that?
> >
> > Yes, that is right. However, if we were to run the experiment for longer we would eventually see the adaptive agent reach the optimal policy first, followed by the Fixed interest agent. The plot is to emphasize the gains in sample efficiency we get from an adaptive interest, however we will also provide the plot for more iterations that show that all agents converge, as this is guaranteed in the tabular case.
> >
> > > In figure 2, how did you compute the MSVE?
> > Here is the equation adapted from Ghiassian et al., 2021
> >
> > $MSVE = \frac{\sum_s d_b(s) i(s) (V^\pi(s) - \hat{V}^\pi(s;\theta) )^2}{ \sum_s d_b(s) i(s)}$
> >
> > Where $d_b$ is the stationary distribution of the behavior policy, i is the interest function, $V^{\pi}$ is the true value function and $\hat{V}^{\pi}$ is the approximate one. We will update the paper to show how MSVE is computed.
> >
> > > Isn't figure 2 showing that the proposed method is slightly worse than the ETDLB method with the TD error as the interest in the tested problem (c.f. subfigure c)?
> >
> > Yes, that is totally right. We wanted to show experiments where a simple heuristic can also work well. The argument we want to make (with Figure 2) is that adapting the interest is beneficial and that there could be many ways to do this. However, we also make the argument that meta gradients are key to obtain gains in the complex control environments we have investigated. Adapting the interest using the TD error (and many more heuristics that we have tried) does not seem to improve the performance beyond this simple environment (FourRooms domains).
> >
> > > I didn't find in the paper a description of the off-policy policy evaluation algorithm (the one you used to produce figure 2).
> >
> > Thank you for pointing this out, we will add the algorithm in the appendix. Those descriptions are available in Appendix A of Ghiassian et al 2021, but we will make sure to have them handy in the paper.
> >
> >
> > > It seems that the interest function is shared across all sub-tasks? If so, why is it reasonably to share it? Why do you consider the multi-task setting instead of the single one?
> >
> > We would like to emphasize that we borrowed the exact same setting as Ghiassian et al. 2021 and this environment is also used in Sutton et al. 2000. Let us clarify and explain the setting here. There are multiple tasks; however, they are disconnected, i.e. there is no sharing of knowledge between tasks. That means that the agent uses a function approximator for each of the tasks independently. Each of the rooms contains two tasks, that is, reaching each of the hallways. As each of the rooms is slightly different in shape and that the tiles from tile coding (the function approximation) do not align the same, the problem is slightly different for each task.  We will also update the paper to clarify this.
> >
> >
> > > Is your off-policy policy evaluation algorithm a batch algorithm? Are TDRC, Vtrace, and ETDLB online algorithms or batch algorithms?
> >
> > No, all these off-policy algorithms are in their online version and we didn’t use batch algorithms in the paper.

---

> > > ### Author Response · Authors · 2022-08-02
> > > **Response to Reviewer xj8y - Part 3**
> > >
> > > > In figure 3, I see that during the entire training period, cells near the hallway cells have high interests. I can also see that, at the end of training, the further a state is from the hallway state, the lower interest the state is assigned. The paper explains that "as the target policy does not visit them often and not many states bootstrap from them, it is less important for them to be accurate". This is a reasonable explanation. But given that the pattern looks pretty close to the value function, it can also be hypothesized that higher state value results in higher interest. If a state has a high value but is not frequently visited by the target policy, would the interest be high?
> > >
> > > That is an interesting question. A first step to answering would be to distinguish whether the high value comes from the true value function or from the current agent’s value function, which may be inaccurate. If it is the true value function, we could imagine that this information may be useful to propagate the TD error more efficiently in the 4R8T domain, but perhaps it could also struggle in the H4R8T one due to the high variance state not having a particularly high value itself. Overall it wouldn’t always be true across domains. If the value function used is the agent’s current value, it would most likely not be effective as the value itself needs to first be high before it can be used to speed up learning. We have actually tried that heuristic in the linear function approximation domains and it didn’t seem to help significantly, just as it doesn’t help too much in the control experiments. That being said the question leads to some interesting follow-ups such as, could we use another statistic than the mean of the expected future return (the value function), but perhaps instead use its variance to define the interest function. We would hope that those kinds of questions could be investigated in future work.
> > >
> > > That being said, the generality of meta gradients is especially useful here in the sense that the meta learning process is going to learn whatever interest that is most conducive to improving the policy/predictions. The best interest may be problem-dependent, and even time-dependent, so adapting the interest through meta gradients give the possibility to assign a high value to a certain state if it is the right thing to do.
> > >
> > > > What is the on-policy control algorithm?
> > >
> > > We use Proximal Policy Optimization (PPO) as our on-policy control algorithm, which is arguably near on-policy. We will clarify this in the paper.
> > >
> > > > When comparing with PPO and the Meta Learned Target algorithm, do you use the same amount of buffer to store past experience and the same amount of computation?
> > >
> > > Yes, we use the exact same amount of buffer for both algorithms and the same number of training iterations.
> > >
> > >
> > >
> > >
> > >
> > >
> > >
> > > [Zhang et al. 2019]: Shangtong Zhang, Wendelin Boehmer, Shimon Whiteson. Generalized Off-Policy Actor-Critic, NeurIPS 2019.
> > >
> > > [Ghiassian et al., 2021]: Sina Ghiassian, Richard S. Sutton. An Empirical Comparison of Off-policy Prediction Learning Algorithms in the Four Rooms Environment, 2021.
> > >
> > > [Mahmood et al. 2015]: A. Rupam Mahmood, Huizhen Yu, Martha White, Richard S. Sutton. Emphatic temporal-difference learning. CoRR, abs/1507.01569, 2015.
> > >
> > > [Sutton et al. 2000]: Richard S.Sutton, Doina Precup, Satinder Singh. Between MDPs and semi-MDPs: A framework for temporal abstraction in reinforcement learning, 2000.

---

> > ### Comment · Reviewer_xj8y · 2022-08-07
> > **Reply to your response.**
> >
> > Most of my questions were addressed, which is good. But I still feel that the key question, how different interest functions affect learning, was addressed in a vague way. I would like to see a more concrete explanation. So maybe I should rephrase my question to be "how does an algorithm possibly accelerate its learning by adapting its interest function?". For example, consider your motivating example. Is it possible for any learning algorithm to know that the interest function should bias the left states over the right ones, before getting any real experience? If this is not possible, what can we learn from the motivating example? I think a simple example showing how your algorithm learns faster than one without an adaptive interest function would be extremely helpful to me.
> >
> > Several other questions.
> >
> > 1. Problem setting:
> >
> > In the prediction case, is the interest function part of the problem or part of a solution? MSVE depends on the interest function and is part of the problem specification, while the adaptive interest function is part of the solution.
> >
> > 2. "The interest function is a possible solution as it can allocate the computational capacity of the agent towards certain states ..."
> >
> > Is the ALLOCATION OF COMPUTATIONAL CAPACITY being changed with an interest function? Do you change the number of times updating of states with the interest function (in the prediction case for simplicity)?
> >
> > 3. Your algorithm
> > Why do you call it a meta-learning algorithm? Do you really have an outer LOOP and an inner LOOP? Can you combine the two updates to \nu and \eta to obtain a single update?
> >
> > 4. Figure 2
> > If nothing is shared across tasks, why not just have one task?

---

> > > ### Author Response · Authors · 2022-08-08
> > > **Response to the reply to our response - Part I**
> > >
> > > We thank the reviewer for reading our rebuttal and we are glad our responses address the majority of your concerns. We hope our below responses address your remaining questions and we kindly ask the reviewer to reflect it with an updated score.
> > >
> > > > "how does an algorithm possibly accelerate its learning by adapting its interest function?"
> > >
> > > One of the main motivations for emphatic algorithms is to change the effective state distribution by selectively emphasizing and de-emphasizing updates with the interest function (Mahmood et al. 2015). However, using fixed interest functions will limit the effectiveness of this method as it treats each update to a certain state in the same way. On the other hand, adaptive interest functions can selectively modulate the impact of each update through the learning process to account for inaccuracies in the target, changes in the agent’s policy or changes in the environment.
> > >
> > > Importantly, unless the agent knows a priori how to set the interest function (i.e. through an oracle), it has only one option: to adapt it online through experience. This is precisely what we suggest in this work.
> > >
> > > Let's use experiments on linear function approximation on the H4R8T domain to address your question as it provides a good example about the effectiveness of an adaptive interest function. In Figure 3b we see that one of the states being highlighted in the middle of training is the state with high variance (shown in blue in Figure 2d). This is exactly what we would expect from an agent adapting effectively its interest: as this state exhibits high variance it requires more resources to be learned precisely than the states surrounding it. For this reason, the interest is higher in that state, and lower for the neighbours, which means that the updates made in that state have a higher impact on the agent’s parameters. Deciding the impact of each update is important as some samples are more informative than others. Here we approach this by considering a state-dependent function, the interest, to balance this trade-off.
> > >
> > >
> > >  > what can we learn from the motivating example?
> > >
> > > The motivating example in the paper is meant to generally introduce emphatic methods to readers not familiar with the literature and also hint at the interesting possibility of adapting the interest function. We do not try to make an argument that the adaptive strategy in this example is expected to be possible in practice without an oracle, but this is done to try to provide a simple picture from which we can expand with more concrete/practical applications.
> > >
> > > In general, non-emphatic RL updates in the episodic setting will bias towards the first state of the episode (Thomas, 2014). This is sometimes useful, but is not always the optimal choice. The interest function allows us this additional flexibility to specify the relative importance of each state.
> > >
> > > > In the prediction case, is the interest function part of the problem or part of a solution? MSVE depends on the interest function and is part of the problem specification, while the adaptive interest function is part of the solution.
> > >
> > > That is a good question, when the MSVE is being calculated, we do not use the interest function we have learned, as in a sense it would mean that the agent controls where we will evaluate its performance. The interest in the MSVE is part of the problem definition, even though we don’t use it for training.
> > >
> > > There is an interesting analogy with the discount factor. Often, we may use a problem-specified discount for evaluation ($\gamma_{eval}$), but that does not mean we should use the same discount factor for training ($\gamma_{train}$), nor does it mean that using $\gamma_{eval} = \gamma_{train}$ would lead to the best result when evaluating with $\gamma_{eval}$. (Jiang et al., 2015). A similar argument could be made for the interest function.
> > >
> > > The interest used in the MSVE follows from the setting of Ghiassian et al. 2021: we use a room-dependent interest function. More precisely, the interest is 1 for all the states in the room connecting to the hallway. This ensures that “that prediction errors from states outside of a room do not contribute to the error computed for each sub-task”, to quote directly from Ghiassian et al. 2021. Notice that this is done for all the algorithms, even those not using emphatic traces in the solution methods.
> > >
> > > > Is the ALLOCATION OF COMPUTATIONAL CAPACITY being changed with an interest function? Do you change the number of times updating of states with the interest function (in the prediction case for simplicity)?
> > >
> > > By computational capacity we mean the capacity to compute accurately the true value function, in this sense the interest function indeed affects it. We do not change the number of times we update a state. The interest function only affects the impact of an update to the agent's parameters by emphasizing it or de-emphasizing it.

---

> > > > ### Author Response · Authors · 2022-08-08
> > > > **Response to the reply to our response - Part II**
> > > >
> > > >
> > > > > Your algorithm Why do you call it a meta-learning algorithm? Do you really have an outer LOOP and an inner LOOP? Can you combine the two updates to \nu and \eta to obtain a single update?
> > > >
> > > > There is indeed an inner loop and an outer loop, and there is no way to combine the two updates simultaneously to obtain a single update, as we first need to obtain the update parameters $\nu’$ to compute the outer loop. We will clarify this further in Algorithm 1 in Appendix A.
> > > >
> > > > > Figure 2 If nothing is shared across tasks, why not just have one task?
> > > >
> > > > The reason for this would be that each room has a slight difference in topology (shape), leading to slightly different problems. The agent is also free to roam in each of the rooms, which also affects the collected data and therefore the number of updates each room is going to see.
> > > >
> > > >
> > > >
> > > > [Jiang et al., 2015]: The Dependence of Effective Planning Horizon on Model Accuracy. AAMAS 2015.
> > > >
> > > > [Thomas, Degris, 2014]: Philip Thomas. Bias in natural actor–critic algorithms. ICML 2014.
> > > >
> > > > [Mahmood et al. 2015]: A. Rupam Mahmood, Huizhen Yu, Martha White, Richard S. Sutton. Emphatic temporal-difference learning. CoRR, abs/1507.01569, 2015.
> > > >
> > > > [Ghiassian et al., 2021]: Sina Ghiassian, Richard S. Sutton. An Empirical Comparison of Off-policy Prediction Learning Algorithms in the Four Rooms Environment, 2021.

---

### Official Review · Reviewer_ndyW · 2022-07-12

**Rating:** 7
**Confidence:** 3
**Soundness:** 3 good
**Presentation:** 3 good
**Contribution:** 3 good

**Summary:**

The paper proposes a method to learn interest state function via meta-gradients in order to do emphatic RL. This interest function essentially simplifies the credit assignment problem and highlights 'more important / interesting' states.

The authors study their method in multiple settings: toy domain, continuous control and discrete domains as well as in the transfer learning setting. The signal is quite clear, the method which authors propose seems to always provide the benefit compared to standard RL approaches.

The authors study different ways of constructing interest function and provide informative ablations over these, with somewhat limited explanations.

Overall, the paper is clearly written, the ideas are easy to understand and the results look good.

**Questions:**

* How does the TD error interest method perform on MinAtar?
* Why do the authors claim that meta-gradient performs better on Figure 2, where it is clear that TD error interest converges faster and to a better value?

**Ethics Review Area:**

["I don’t know"]

**Limitations:**

Limitations are clear from the paper

**Strengths And Weaknesses:**

Strengths:
* Clearly written paper
* The authors studied a novel approach for RL and credit assignment via meta-learning of the interest function and provided solid experimental evidence that the approach can improve RL
* The experiments are quite broad, demonstrating the applicability of the method in different scenarios: discrete control, continuous control, transfer learning.

Weaknesses:
* It is not clear why the authors didn't provide `TD error interest` for MinAtar. Given the results on the toy domain, this approach may actually perform better than meta-gradients.
* Minor writing / visualization details. The legend colours on the Figure 2 are very hard to read.

---

> ### Author Response · Authors · 2022-08-02
> **Response to Reviewer ndyW**
>
> Thank you for your valuable feedback and comments. Please let us know if there are more questions.
>
>
> > How does the TD error interest method perform on MinAtar?
>
> Thanks for your suggestion. We ran new experiments with TD error interest on MinAtar and provided new results for this question in the following link: https://imgur.com/pP1ZzJT . We have also updated the paper with these new results (please see Figure 4). These results show that, as in the MuJoCo experiments, the TD error interest baseline does not perform well in control, which is in contrast to the results in policy evaluation. This also highlights that meta-gradients are more flexible and more adaptable to the specific setting.
>
>
> > Why do the authors claim that meta-gradient performs better on Figure 2, where it is clear that TD error interest converges faster and to a better value?
>
> That is a good question and thanks for pointing this out. In section 4.1, we try to make the argument that an adaptive interest, whether it is adapted through meta-gradients or through the TD error, provides improvements in terms of performance. We do not try to make the argument (in Figure 2) that meta-gradients are always the best choice and for that reason we provided this additional simple heuristic that performs as well and sometimes better in the linear function approximation setting. That being said, this simple heuristic doesn’t seem to scale to the control setting in section 4.2. We will make this clear by highlighting that both baselines, meta gradients and the TD error, are adaptive by making the plots more explicit. In this work our general argument is that adapting the interest is a promising area of research and that there may be different choices that work well, an effective and general one being meta-gradients.

---

> > ### Comment · Reviewer_ndyW · 2022-08-09
> > **Thank you for the response**
> >
> > Thank you for the answer and for adding results to the paper.
> >
> > >  In this work our general argument is that adapting the interest is a promising area of research and that there may be different choices that work well, an effective and general one being meta-gradients.
> >
> > I think this makes sense, thank you for clarifying.
> >
> > I will keep my score after this response.

---

> > > ### Author Response · Authors · 2022-08-09
> > > **Thank you**
> > >
> > > Thanks again for suggesting those experiments, comments, and responses. Please let us know if you have more questions or comments.

---

### Author Response · Authors · 2022-08-02
**Response to all reviewers**

We thank the reviewers for their useful feedback that we've used to greatly improve the paper. We have responded to the concerns of the reviewers as individual comments below.

All reviewers agree that our paper is well-written, well-motivated, our experiments are comprehensive, and our method is simple/elegant/straightforward. To summarize our contributions: i) we propose a simple but effective way to learn and adapt the interest function using meta-gradients and show that adapting the interest function from a stream of data consistently leads to improved performance. ii) Our results on a wide range of different settings, further highlight the value of our proposed method compared to other methods.

Per Reviewer ndyW’s and UBwo's suggestion, we also ran "TD error interest" method for MinAtar experiments to further situate our method relative to this approach. This result provides another data point that our method is an effective approach to learn interest function. Please see our responses below where we discuss this new experiment.

We also made minor updates to the paper and appendix to address some of the reviewers’ comments. Please see our responses below that mention what has been updated in the text.

---

### Meta-Review · Area_Chair_YzjJ · 2022-08-24

**Recommendation:** Accept
**Confidence:** Certain

**Metareview:**

Well-written and interesting paper that meta-learns the interest function in emphatic RL, rather than using a fixed interest function. The idea is well-motivated and a comprehensive empirical study is performed. There was an an active discussion among the reviewers and authors, in which the authors addressed the various reviewer questions effectively and which resulted in an updated presentation and an increase in scores. Overall, a clear accept.

**Award:**

No

---

### Decision · Program_Chairs · 2022-09-14

Accept